# Mechanobiologically-optimized non-resorbable artificial bone for patient-matched scaffold-guided bone regeneration

Jonathan R. Clark[1,2,3,4,5] ✉, D. S. Abdullah Al Maruf[2,3], Eva Tomaskovic-Crook[6,7,8], Kai Cheng [2,4], William T. Lewin [6,8,9], Hai Xin [2], Boyang Wan[1,10], Jiongyu Ren[1,11], Chi Wu[10], Hedi V. Kruse[6,9,12], Daniel K. Lawrence[13], Innes Wise[14], Aditi Gupta[1,2], Maria A. Woodruff[1,15], Maryam Alsadat Rad[6,8], David Leinkram [2,3,5], Timothy Manzie [1,2,3,5], Krishnan Parthasarthi[2,3,5], James Wykes[2,3,5], Tsu-Hui Hubert Low[2,3,16], Dale Howes[2,17], Catriona Froggatt [2,5], Ruta Gupta[1,3,18], Gordon Wallace [1,7], Qing Li [1,10,19], David R. McKenzie [6,9,12] & Jeremy M. Crook [1,6,7,8,9] ✉

Scaffold-guided bone regeneration is poised to revolutionize the management of critical-sized bone defects. However, translation into clinical practice has been hampered by the focus on bioresorbable scaffolds where the rate of degradation needs to match the rate of bone formation and metal plates are required to overcome their mechanical limitations. Metal plates are problematic because they cause stress shielding and X-ray perturbation, increasing the likelihood of hardware failure and interfering with post-operative radiotherapy and imaging. Segmental defects of the mandible are challenging due to high tensile and shear stresses encountered during mastication, with the ovine mandible especially vexing because of the high repetitive loads. Here we show long-term reconstruction of ovine segmental mandibulectomy defects using a permanent, patient-matched, numerically optimized, 3D-printed, thermally toughened, plasma-treated, and laser-sintered polyetherketone gyroid scaffold housing a resorbable ceramic lattice infused with a stem cell laden hydrogel serving as an osteoinductive reservoir of calcium. The durable clinical performance observed indicates a translatable alternative to traditional reconstruction using bone grafts with metal plate fixation.

Critical-sized bone defects represent a significant global health challenge. It is estimated that more than 1.6 million bone grafts are used to treat such defects in the U.S. each year, costing a staggering USD$244 billion[1,2]. Whilst critical-sized bone defects can occur anywhere in the body from a variety of causes, segmental defects of the mandible created from excision of bone in the treatment of cancer are one of the most challenging to manage due the high aesthetic and functional importance of the jaw, the extreme repetitive tensile and shear stresses associated with mastication, the frequent use of radiotherapy, and the

need for prolonged surveillance imaging[3,4]. Solving this challenge in the 'worst case' will have widespread implications for the use of scaffold-guided bone regeneration (SGBR) in other locations and etiologies.

Large segmental defects are traditionally reconstructed using vascularised autologous bone grafts taken from sites such as the fibula, pelvis, or scapula and fixated using plates made of titanium (Ti), Ti alloys (e.g., Ti-6Al-4V), or other metallic materials such as austenitic grade stainless steel (e.g., SS316L)[5,6]. Metal plate fixation is essential

until the grafted bone has united, which may fail to occur in up to 60% of cases[7]. Unfortunately, bone grafted from anatomically disparate sites is unable to replicate the three-dimensional (3D) structure of the native bone and there is substantial morbidity inherent in these complex reconstructive procedures. This morbidity includes the sacrifice of muscles, nerves, and blood vessels during bone harvest and is exacerbated by subsequent complications of metal plate fixation, requiring plate removal in ~25% of patients[8–10]. The high elastic modulus of Ti and other metals causes both mechanical stress shielding and unwanted stress concentrations, which may contribute to bone resorption and osteolysis, impairing bone healing and increasing the likelihood of fixation hardware failure[11]. The stress shielding associated with high-modulus materials may be lessened by 'stiffness matching' which can be achieved by changing material composition through the use of composites or by changing its geometry, i.e., location, shape or porosity[12–14]. Unfortunately, this is not always feasible due to anatomical constraints and whilst some problems may be overcome by optimizing the mechanical properties, this does not address issues related to X-ray perturbation during postoperative radiotherapy and associated imaging artifacts. X-ray perturbation can increase the dose of radiotherapy to healthy bone, under-dose the target volume, and interfere with surveillance imaging modalities, impeding early detection of infection and local recurrence of cancer (Fig. 1)[15–17].

There is considerable interest in using SGBR in the form of patient-matched medical devices (PMMDs) that are additively manufactured to precisely match the defect size and shape and abrogate the need for autologous bone grafts. Despite this, SGBR has failed to deliver constructs that are suitable for use in clinical practice because most tissue engineering strategies employ bioresorbable scaffolds that are progressively replaced by bone as they degrade[11,18–21]. Examples of biomaterials trialed for reconstructing segmental bone defects in large animal models include ceramics such as hydroxyapatite and beta-

tricalcium phosphate (βTCP) and various polymers (poly-caprolactone, poly-lactic acid, poly-lactic-co-glycolic acid, polyurethane and hydrogels), in combination with growth factors and cellular components. This concept is attractive; however, it relies (unrealistically) on synchronizing the rates of scaffold degradation and osteogenesis to achieve the desired biomechanical properties in the medium- to long-term. Furthermore, whilst most bioresorbable scaffolds are designed to withstand compressive loads, they require reinforcement with metal plates or intramedullary nails to address issues with fixation, bending moment, and torsional stability. In fact, there are neither preclinical large-animal studies nor clinical studies published where SGBR has been used to reconstruct segmental bone defects without metal plate augmentation, and most studies reconstruct bone defects that are too small to be relevant to oncology, which are typically 6 cm or more. The most promising 3D-printed biomaterial for SGBR thus far appears to be a combination of polycaprolactone (PCL) and βTCP, which can be fabricated as a porous biocompatible construct with suitable structural and osteoconductive properties. However, even with rigid fixation, all bioresorbable scaffolds will ultimately fail if the rate of degradation exceeds the rate of new bone formation[21–23].

In contrast, an additively manufactured fracture-tough, permanent, and biocompatible high-performance polymer scaffold with a bone-like elastic modulus has several advantages over bioresorbable and complex biomimetic scaffolds, including more predictable and tunable mechanical properties for optimal load bearing and osteoconduction. The polyaryletherketone (PAEK) family of polymers fulfills many of the requisites for SGBR, inclusive of oncological indications, with a proven safety profile and radiolucency to avoid the issues related to X-ray perturbation[11]. The most widely known members, polyetheretherketone (PEEK) and polyetherketoneketone (PEKK), have been employed in craniofacial implants for decades[24]. However, as hydrophobic and bioinert polymers, they lack the osteoconductive

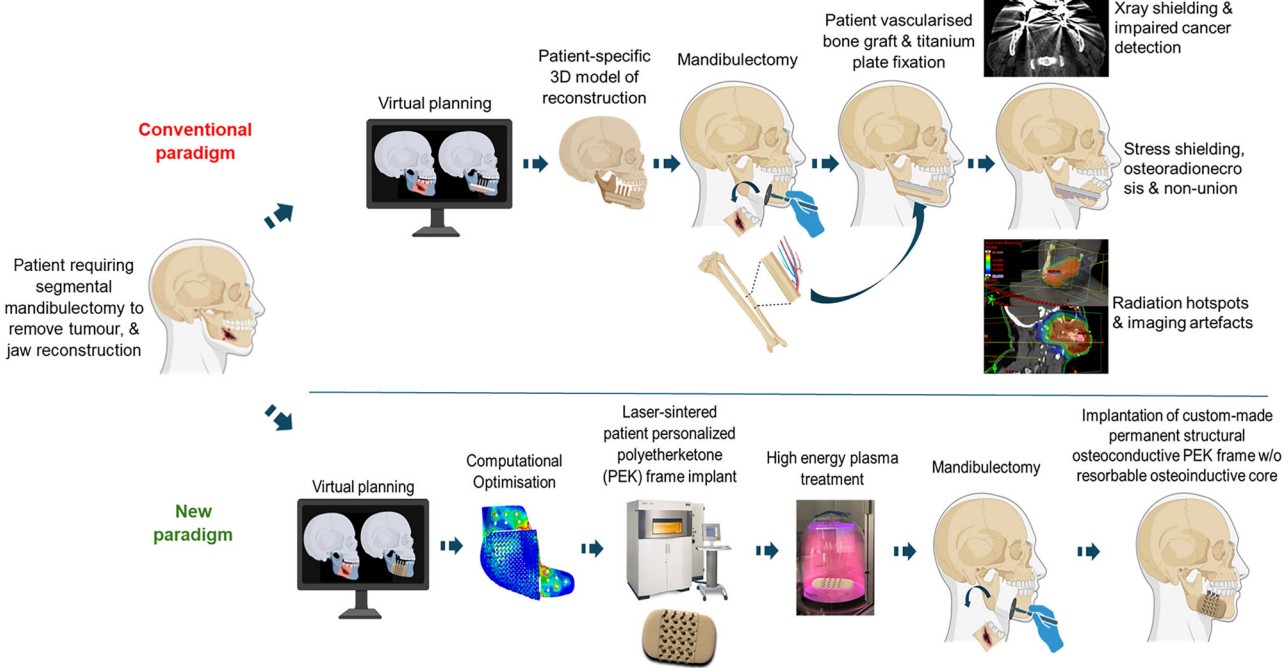

**Fig. 1 | A new paradigm in patient-matched scaffold-guided bone regeneration.** Current clinical practice (Conventional paradigm) uses vascularized autologous bone grafts that are transplanted from anatomically disparate sites to be reshaped and fixated with Ti plates. This approach is associated with considerable morbidity from tissue harvest, and the suboptimal material properties of Ti increase the risk of non-union and impair radiological surveillance. The new paradigm uses 3D-printed artificial bone (LS-PEK) that is custom-made for the individual's defect site, numerically optimized for osteoconduction, thermally toughened for strength, and surface-treated for osseointegration, thus avoiding the morbidity of bone grafts and the need for metal plate augmentation. Created in BioRender. Crook, J. (https://BioRender.com/e8aq7lf).

properties of Ti and ceramics, and so fail to osseointegrate[25]. We have previously shown that PEEK can be surface-modified to increase its bioactivity and osseointegration using nitrogen plasma-immersion ion implantation (PIII)[25–28]. In nitrogen PIII, the polymer material is immersed in nitrogen plasma (ionized nitrogen gas) and subjected to a high-voltage pulse (-10 keV) causing positively charged nitrogen ions to bombard and become embedded within the polymer surface. Free radicals created by the plasma treatment process form covalent bonds with adjacent proteins and substantially increase hydrophilicity and subsequent cell attachment and tissue infiltration. A less well-known member of the PAEK family, polyetherketone (PEK), is a high-performance semi-crystalline thermoplastic that is similarly fracture-tough, low-fatigue, radiolucent, able to be PIII treated, sterilizable, and has an elastic modulus better matching bone, however, there are no publications of its use in segmental bone defect repair[29].

Here we describe an alternate paradigm in patient-matched SGBR using PEK in the form of a nitrogen PIII-treated thermally toughened frame with a gyroid-based triply periodic minimal surface (TPMS) scaffold structure, and an osteoinductive bioresorbable stem cell-infused ceramic core. Until recently, additive manufacturing techniques used on PEEK/PEKK, such as fused filament fabrication (FFF), have been unable to meet the desired print resolution or strength requirements for load-bearing implants. In contrast, PEK can be additively manufactured through the process of powder bed fusion using laser sintering (PBF-LS), which is commonly known as selective laser sintering (LS), to achieve the more complex and intricate geometries required for SGBR. LS-PEK can be toughened through annealing for improved mechanical performance and the micro-rough surface topology created through LS promotes tissue integration[30]. The artificial bone is custom-designed using virtual surgical planning (VSP) to match the defect site, mechanically optimized using image-based finite element (FE) analysis for strength, and numerically modelled to promote bone ingrowth. The mechanical optimization of LS-PEK provides sufficient structural stability to abrogate the need for both metal plate augmentation and autologous bone grafts. The LS-PEK frame is then dry-ice blasted for cleaning, annealed, and nitrogen PIII-treated, the latter able to withstand heat sterilization (Fig. 1).

It is conceivable that by engineering complex biomimetic structures containing multiple tissue lineages (bone, muscle, and vasculature), a more functional construct could be deployed[31–34]. Riffai et al. define the "Quad of tissue engineering", comprising biomaterials, regenerative cells, morphogens/cytokines, and fabrication modality, each considered integral to repairing complex tissues such as bone. They suggest that bioink combinations with 3D-printed PCL may be structurally suitable for bone regeneration but at the same time highlighting the lack of clinically relevant scaffolds employing bioinks[35]. However, additional complexity encumbers clinical translation due to regulatory constraints, greater unpredictability, and the unsolved problem of how the viability of large living structures with complex geometries can be maintained when implanted. At present, this concept is better suited to tissue-engineered models for drug development. Furthermore, it is challenging to incorporate complex structures such as blood vessels into biomaterials after they have been 3D-printed into scaffolds. This makes co-printing biomaterials with bioinks the most intuitive solution. Unfortunately, the requirement for bone scaffolds to be mechanically appropriate restricts the range of biomaterials where this is feasible with many requiring high temperatures for printing or sintering that is incompatible with living tissues. To overcome this challenge, we incorporated an osteoinductive bioresorbable core that was placed within the LS-PEK frame at the time of surgery. Importantly, the ceramic core was composed of 3D-printed βTCP, designed to serve as a calcium reservoir, but non-essential for structural stability. This was infilled with gelatine methacryloyl (GelMA) encapsulating adipose derived stem cells (ADSCs) that were osteogenically pre-differentiated prior to assembly to enhance the

bioavailability of βTCP. Finally, we tested our LS-PEK implant in a mature ovine segmental mandibulectomy model. Most SGBR studies employ in vivo models where the loads are considerably lower than in humans, e.g., small animals or quadruped long bones[36]. In contrast, the mature ovine segmental mandibulectomy model is a 'worst case' scenario, exaggerating the mechanical and biological challenges of SGBR in humans. Importantly, all sheep demonstrated normal masticatory function due to osseointegration of the PIII-treated LS-PEK implant with progressive stress-driven osteoconduction through the scaffold.

## Results

### Material selection and surface treatment to enhance osseointegration, osteoconduction, and osteoinduction

Three independent mature ovine studies were performed to evaluate candidate biomaterials for the artificial bone (Fig. 2). For the first study (phase 1), the bone-implant interface of PIII-treated LS-PEK sawtooth cylinders was compared with replicate grade 23 Ti cylinders manufactured by powder bed fusion using laser beam (PBF-LB) that were implanted into the mandible for 8 to 12 weeks. Histomorphometry showed that bone implant contact was equivalent in the 0-24 μm and 24-80 μm regions, indicating that PIII-treated LS-PEK has similar osseointegration to PBF-LB-Ti, the standard biomaterial in clinical use[37]. For the second study (phase 2), PIII-treated LS-PEK and PIII-treated FFF-PEEK scaffolds employing a Schwartz P TPMS architecture were implanted into critical-sized (2 cm) partial bone defects created in the mandible for 13 weeks[38]. In vivo osteoconduction and osseointegration of both candidates were compared using μCT and tensile testing. New bone volume and pull-out force were greater in explanted PIII-treated LS-PEK scaffolds due to the higher surface roughness, enhancing osseointegration and osteoconduction. For the third study (phase 3), the osteoinductive potential of various calcium phosphate substitutes, hydrogels, and stem cells (autologous and allogeneic ADSCs) selected from previous studies[39,40] were compared with autologous bone and platelet rich fibrin using an in vivo PIII-treated LS-PEK bioreactor model implanted in the sheep scapula for 12–16 weeks. This model allowed simultaneous contact between multiple samples of biomaterials (PCL, GelMA, βTCP, commercial calcium phosphate substitutes (BioOss®, Cerabone®, Zengro®), autologous bone graft, and GelMA encapsulated autologous and allogeneic ADSCs) and both periosteum and cortical bone in each sheep[41]. Endochondral ossification was observed in bioreactors containing βTCP and ADSC-laden GelMA (Fig. 2C); the predominant osteogenesis pathway in hypoxic environments and a desirable property in SGBR.

### Mechanobiological design of a patient-matched artificial bone for SGBR

We designed an artificial bone implant for reconstructing critical-sized segmental bone defects following the sequence shown in Fig. 3 and Supplementary Movie 1.

The artificial bone has three components: An annealed and PIII-treated LS-PEK frame with a TPMS internal architecture, an osteoinductive selectively polymerized βTCP lattice core infilled with ADSC-laden GelMA, and a LS-PEK crossbar used to secure the βTCP lattice (Fig. 4). The LS-PEK frame was custom designed for each sheep's mandibular defect using a clinically validated VSP protocol[5]. The VSP process commenced with the acquisition and segmentation of sheep-specific DICOM data from high resolution CBCT scans and the generation of 3D digital stereolithography (STL) models of the mandible. The LS-PEK frame bridging the 6 cm bone defect employed a single gyroid TPMS architecture (Fig. 4A). The βTCP lattice forming the core of the artificial bone bridged the defect to contact native mandibular bone at each end, serving as a local reservoir of calcium. An open lattice design facilitated GelMA infusion and transmission of UV light through the depth of the structure to ensure adequate crosslinking of GelMA.

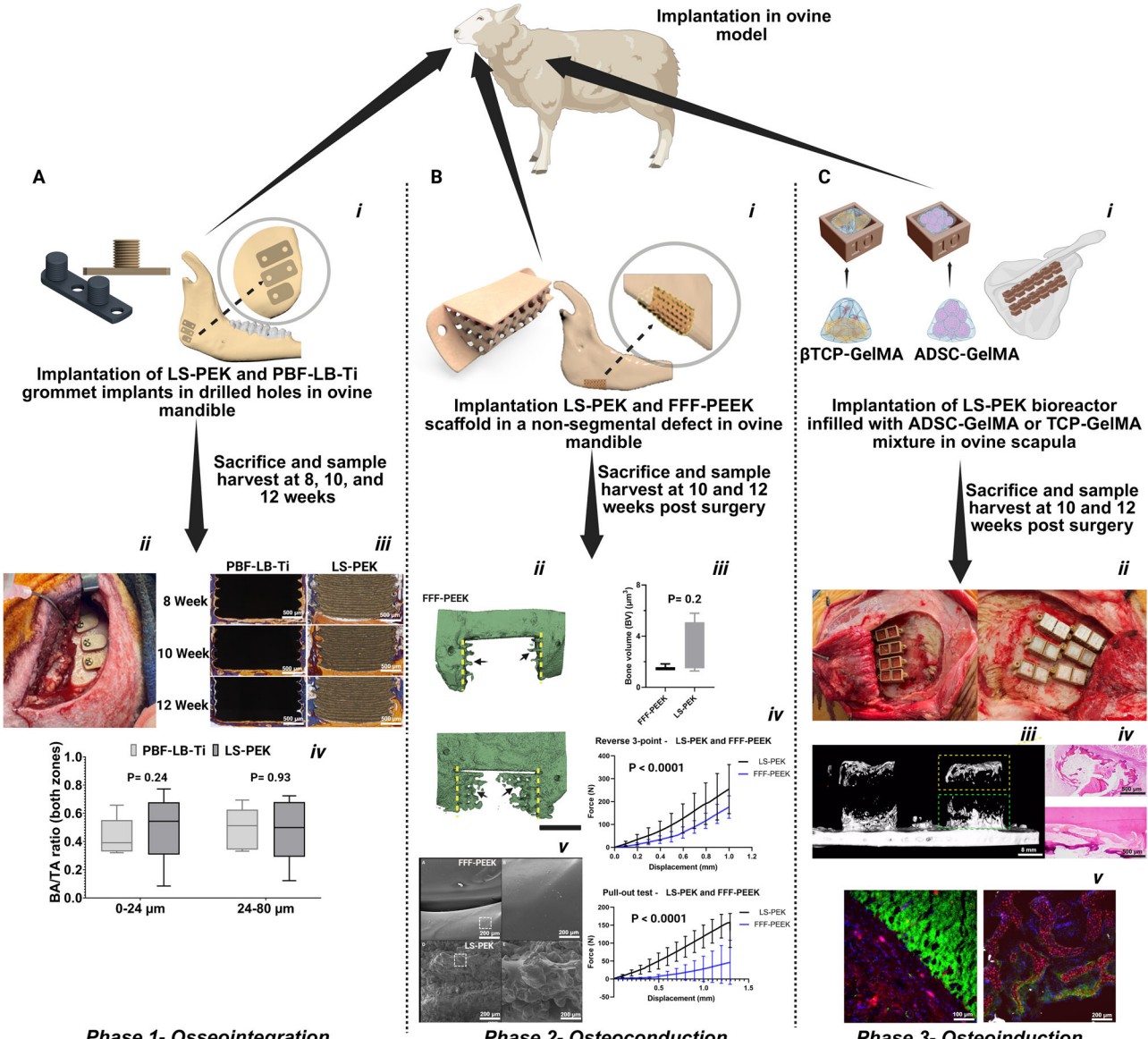

**Fig. 2 | PIII-treated LS-PEK demonstrated similar osseointegration to implant-grade PBF-LB-Ti and superior osteoconduction to PIII-treated FFF-PEEK.** βTCP and allogeneic ADSC-laden GelMA demonstrated endochondral ossification. **A** Phase 1: i-ii. PIII-treated LS-PEK and PBF-LB-Ti cylinders were implanted in sheep mandibles. iii. Resin-embedded histomorphometry using Goldner's trichrome stain (blue = bone, yellow = soft tissue; scale bar, 500 μm). iv. Box-and-whisker plots (box: mean, 25th–75th percentile; whiskers: min–max) show similar bone area (BA)/total area (TA) contact at 0–24 μm ($p = 0.24$) and 24–80 μm ($p = 0.93$) from the implant ($n = 5$; repeated measures ANOVA, $F(1.032, 4.130) = 0.163$, $p = 0.71$), with paired two-tailed t test comparing zones yielding $p = 0.5134$. **B** Phase 2: i. Non-segmental mandibular defects were reconstructed with PIII-treated FFF-PEEK or LS-PEK scaffolds with a solid partition. ii. μCT 3D reconstructions demonstrate new bone formation (yellow dashed lines = osteotomy margins; black arrows = new bone; scale bar, 34 mm). iii. Comparing bone volumes showed no significant difference between FFF-PEEK (median 1.460, $n = 3$) and LS-PEK (median 2.560, $n = 4$; one-tailed Mann–Whitney U, $p = 0.2000$; mean ± SEM). iv. Reverse 3-point bend ($n = 5$) and

pull-out ($n = 3$) testing revealed significant differences in force–displacement profiles (quadratic regression, $F(3,2402) = 319.4$, $p < 0.0001$; $F(3,1944) = 1769$, $p < 0.000$), with LS-PEK achieving maximum force at failure of 228 N at 1.8 mm. Data are mean ± SD. v. SEM revealed higher surface roughness of LS-PEK compared with FFF-PEEK (scale bar, 200 μm). **C** Phase 3: i. Double-layer in vivo bioreactors (upper layer contacts scapular periosteum and the lower layer contacts cortical bone) in sheep, allowing us to evaluate multiple materials (PCL, GelMA, βTCP, CaP substitutes, autologous bone, and GelMA-encapsulated ADSCs). ii. Surgical implantation of scapula bioreactors. iii. μCT 3D reconstructions show new bone formation within radiolucent bioreactors (scale bar, 8 mm). iv. Histology of ADSC–GelMA bioreactors demonstrated endochondral ossification (scale bar, 500 μm; $n = 4$ per layer, 5 sheep total). v. Immunohistochemistry confirmed viable osteogenic cells (GelMA autofluorescence = green; CD44+ bone lineage cells = red; DAPI nuclei = blue; scale bars: left 100 μm, right 200 μm, $n = 4$). Created in BioRender. Crook, J. (https://BioRender.com/0dah66z). Source data are provided as a Source Data file.

During the design phase, in silico analysis of the structural stability was performed using FE modelling and validated with in vitro mechanical testing using cadaveric sheep mandibles (Fig. 4B-H)[4,42]. The von Mises (VM; Fig. 4C) and maximum principal stress (MPS; Fig. 4D) distributions and extended finite element method (XFEM)-simulated crack patterns in the LS-PEK frame (Fig. 4B) were used to modify

regions at risk of failure for each sheep. Here, mechanical testing simulated the planned in vivo implantation by insetting the LS-PEK frame into segmental defects of the full (bilateral) mandible, fixated with bicortical Ti screws, and applying loads that replicate the muscles of mastication without fixing the condyles (Fig. 4B). In these mechanical tests, implant failure occurred at >1500 N, three times the

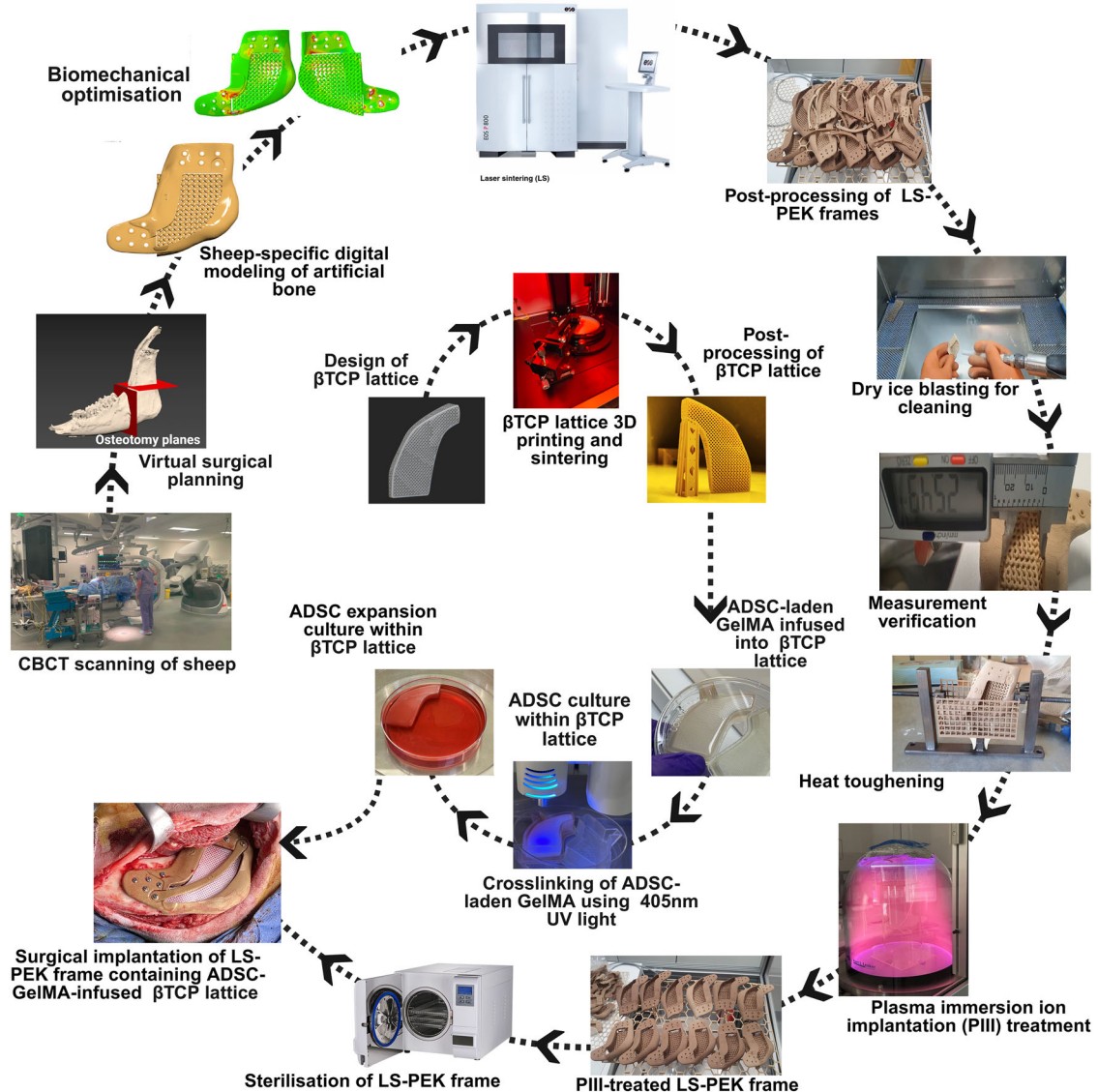

**Fig. 3 | Schematic of the sheep-specific artificial bone design and manufacturing process for reconstructing segmental bone defects of the mandible.** CBCT scans of the sheep mandible were acquired. Virtual surgical planning (VSP) was performed to define each mandibular defect and design the sheep-specific LS-PEK frame and cutting guides. Biomechanical optimization using numerical modelling was performed. The LS-PEK frame and scaffold was manufactured using laser sintering (LS), followed by post-processing, dry ice blasting, dimensional verification, heat toughening, plasma immersion ion implantation (PIII) treatment, and heat sterilisation. In parallel, a βTCP lattice was 3D-printed, furnace sintered, infused with ADSC-laden GelMA, and cultured for ADSC expansion and osteogenic differentiation. Crosslinking of GelMA was performed using 405 nm UV light to stabilize the cell-hydrogel construct. The βTCP lattice was placed in the PIII-treated LS-PEK frame at the time of surgical implantation into the mandibular defect and secured with a LS-PEK crossbar. Created in BioRender. Crook, J. (https://BioRender.com/m66k2rs).

maximal bite force of human premolar intercuspal clenching (450 N)[43,44]. To further optimize the design, a time-dependent mechanobiological model was used to simulate bone growth into the scaffolds using different contact conditions at the interfaces between natural bone, the LS-PEK scaffold, and the βTCP lattice, set to match the biting forces of the sheep (Fig. 4E-H)[45].

## Durable in vivo biomechanical performance in segmental bone defect reconstruction

Sheep were surgically implanted for 6 or 12 months according to the sequence shown in Supplementary Fig. 1 and Fig. 5. Serial CBCTs of the implants showed a progressive increase in new bone volume in all sheep, ranging from 1.10 cm³ to 3.37 cm³ (Fig. 5J-O, Supplementary Fig. 2, Supplementary Fig. 3). Sheep commenced oral intake immediately following recovery from anesthesia (Fig. 5H). Sheep were fed with

a modified chaff and hay diet and exhibited minimal discomfort with no long-term changes in chewing behavior (Supplementary Movie 2). Notably, the mean chewing rate, which serves as an indicator of masticatory function[46], was 120 chews per minute prior to surgery and 129 chews per minute post-surgery, indicating normal masticatory efficiency of the sheep. The key metrics of success were clinical because implant failure, elevated by the high masticatory rate of sheep, will first cause a local inflammatory response and pain. Sheep manifest pain as anorexia and decreased oral intake and rapidly lose weight if oral function and nutrition is impaired[47,48]. All implanted sheep demonstrated similar behavior and weight-gain to non-implanted controls (mean weight gain 3.5 kg, range 0.5 – 7.2 kg; Fig. 5Q). At sacrifice there was no capsule formation, granulation tissue, seroma, or suppuration and the associated masticatory musculature was strongly adherent to the artificial bone during explantation (Supplementary Fig. 4). This

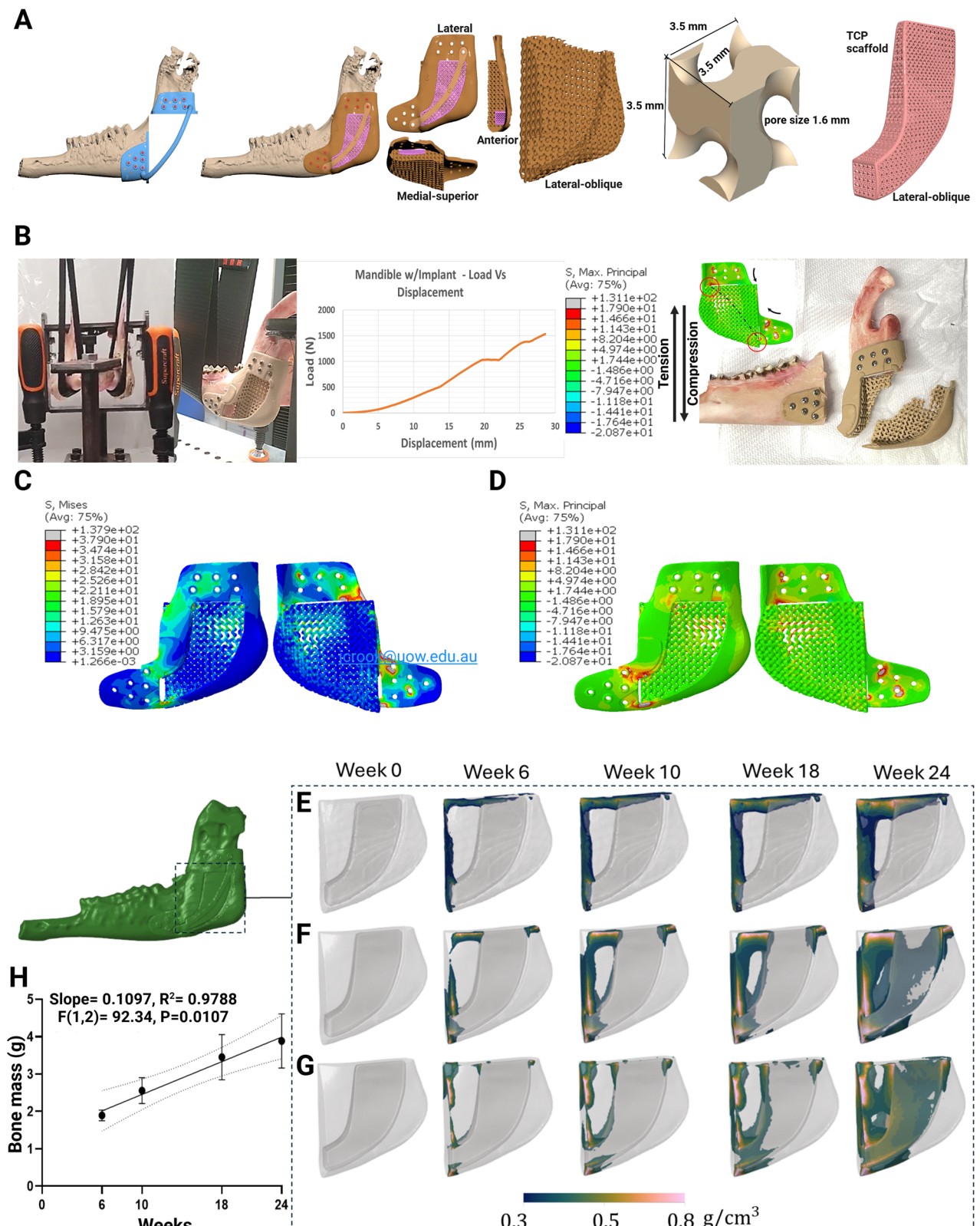

soft-tissue integration was confirmed prior to histological analysis when the adjacent musculature had to be excised from the scaffold prior to mechanical testing and resin embedding. The clinical and macroscopic evidence of implant integration was then confirmed microscopically by the absence of any unwanted inflammatory response on histological assessment as detailed below (Fig. 5I).

**Explant analyses confirm reliable osseointegration despite variable degrees of osteogenesis**

μCT demonstrated osteogenesis through the gyroid structure ranging from 0.93 cm³ to 2.71 cm³, representing 11% to 38% of the available LS-PEK scaffold volume. In some cases, new bone bridged the entire 6 cm bone defect (Fig. 6). As PEK is radiolucent, CT cannot directly assess

**Fig. 4 | Mechanobiological design optimization: Image-based FE analysis with mechanical testing validation, predicted fracture propagation using XFEM, and simulated bone ingrowth over 6 months using time-dependent mechanobiological numerical modelling. A** Schematic of the cutting guide, LS-PEK frame, βTCP lattice, and LS-PEK restraining crossbar customized to live and cadaveric sheep used for design, biomechanical in silico modelling, and mechanical testing (cadaveric only). The single gyroid TPMS scaffold is shown including parameters of the gyroid unit cell. **B** Image-based FE model of whole-mandible and artificial bone and in vitro mechanical testing set-up simulating incisal loading from action of masseter and medial pterygoid musculature. Force-displacement curve from mechanical testing showing implant failure at 1533 N. The LS-PEK implant fractured in the predicted locations during mechanical testing validating the XFEM-predicted crack propagation pattern (black dashed line). **C** von Mises and (**D**) maximum principal stress distributions from image-based FE simulations used to iteratively modify the design of the LS-PEK frame. **E–H** Bone growth mechanobiological simulations over 24 weeks for different contact boundary conditions at the interface between natural bone and the scaffold component of the LS-PEK frame, as well as at the interface between the LS-PEK scaffold and βTCP lattice. **E** Simulation assuming no contribution from the βTCP lattice. **F** Simulation assuming frictional contact conditions of the βTCP lattice. **G** Simulation assuming bonded contact conditions of the βTCP lattice. For **E–G** coloured regions represent newly formed bone with a density above 0.3 g/cm³. Dark blue indicates low-density bone (0.3 g/cm³), transitioning through green and yellow (0.5 g/cm³), to pink, which represents high-density bone (0.8 g/cm³). **H** Mean ( ± SD) bone mass from the three simulations (**E, F**) and **G** The estimated bone mass at week 24 was 3.89 g. Linear regression analysis established a significant relationship between bone growth and time after implantation, $F_{(1,2)} = 92.34$, $p = 0.0107$, with a slope of 0.1097 and R-squared = 0.9788. Linear regression plot presents line of best fit (solid line) and 95% confidence interval (dotted line). Source data are provided as a Source Data file.

the bone-implant interface, however, the voids were filled with new bone interlocking the gyroid structure (Figs. 5U-W, 6A(vii), 6B). Histomorphometry on resin-embedded specimens confirmed excellent bone-implant contact (Fig. 6A), consistent with histological osseointegration of the PIII-treated LS-PEK frame. However, there was less new bone associated with the βTCP lattice core, ranging from 0.08 cm³ to 0.74 cm³, and the degradation rate of the βTCP lattice was highly variable (Figs. 5P, S and 6B). Bone within the lattice was limited to discrete islands except where the lattice contacted the native mandible (Fig. 6Aii-v, vii-ix). Immunofluorescence staining for von Willebrand Factor (vWF), a marker of endothelial cells and blood vessels, revealed strong positive signals in both βTCP (Fig. 6Avi, Supplementary Fig. 5) and LS-PEK (Fig. 6Ax, Supplementary Fig. 5) implant sections. Specifically, intense vWF expression was observed at the interface between the host bone and the newly formed bone surrounding each type of implant. The level of inflammation was assessed by immunohistochemistry using key macrophage markers: CD68 (pan-macrophage marker, ab125212), CD206 (M2-like, anti-inflammatory marker, ab64693), and iNOS (M1-like, pro-inflammatory marker, ab15323). As shown in Supplementary Fig. 6, the macrophage response characterized by these markers demonstrated low overall inflammatory response to the implanted scaffolds. CD68 showed negligible positivity around βTCP implants and weak positivity near LS-PEK implants, indicating a low overall macrophage presence but slightly more pronounced in LS-PEK sections. CD206 staining showed medium positivity around βTCP and low positivity around LS-PEK, suggesting a more prominent M2-like (anti-inflammatory or tissue-remodelling) response to βTCP. Finally, iNOS exhibited very weak positivity around both implant materials, indicating negligible M1-like (pro-inflammatory) macrophage activation.

Explants underwent non-physiological tensile testing of the hemimandible-implant complex where the condyle and coronoid process were fixed in resin to assess osseointegration (as distinct from FE-validatory mechanical testing prior to implantation where the whole mandible was used, and the setup matched physiological loading) (Fig. 7A, B). The fixation screws were removed, and testing ended prior to catastrophic failure to allow histological analysis of the intact bone-implant interface (Fig. 7C). The maximum force applied was 641 N compared with 601 N for the non-implanted hemimandible side ($p = 0.042$). Failure mostly occurred in the native mandible or teeth without separation of the bone implant interface, confirming mechanical osseointegration.

### Osteogenesis-dependent biomechanics

The influence of osseointegration on the mechanical stability of the implanted artificial bone was explored through FE simulations where the predicted displacement from the applied force was output during the loading process (Fig. 7E). There was high conformity between the explanted mechanical tests and the simulated load-displacement curves derived from FE models with bone ingrowth ($R^2 = 0.96$). In the model without osseointegration, the peak VM stress concentration (73 MPa) was observed around the anterior bone-implant interface (Fig. 7Fi). In contrast, a more even stress distribution was seen in the middle of the LS-PEK frame with osseointegration, and more VM stress (65 MPa) was transferred to the posterior mandible, indicating the bone actively shared the occlusal load (Fig. 7Fii). The shift of peak von Mises stress from the implant to the mandibular bone suggests that the implant is bearing less load, thereby reducing the risk of ductile-related damage such as yielding or fatigue failure in the implant. In the model without osseointegration, the region with the highest tensile MPS (95 MPa) was around the screw holes (Fig. 7Gi). However, when osseointegration and bone growth occurred, the load was transferred to the LS-PEK frame through the bone-implant interface, rather than relying on the screws (Fig. 7Gii). Although some concentration of MPS was observed in the upper left re-entrant corner of the scaffold, the majority of the MPS was distributed on the upper surface of the posterior mandible.

### Stress-dependent osteogenesis

Supplementary Movie 3 shows stress-driven CBCT bone growth patterns. By week 6, new bone had formed at the anterior and posterior interfaces of the mature bone and the gyroid scaffold component of the LS-PEK frame where the concentration of osteoblasts was relatively high due to the migration of cells from mature bone areas. By week 10, the newly formed bone had extended into the voids of the scaffold. This was most pronounced in two high-stress areas: the anterior aspect of the posterior bone interface and the inferior aspect of the anterior bone interface. Figure 8 represents the changes in predicted bone mass based on numerical models compared to the in vivo CBCT data in Sheep D over 24 weeks. The in vivo data here best matched the simulation shown in Fig. 4E, which assumed no mechanical contribution from the βTCP lattice, however, the pattern of osteogenesis varied between sheep. At the conclusion of the study, the predicted total bone mass was 1.4% higher than bone mass calculated from μCT data (3.062 g vs 3.020 g; Fig. 8). Whilst the pattern of bone growth followed the simulations, there was marked variability in both the total bone mass and distribution between sheep.

### Discussion

This trial represents essential evidence that patient-matched SGBR using permanent additively manufactured scaffolds is a viable option for reconstructing segmental bone defects without autologous bone grafts or metal plate augmentation. The mature ovine mandibular model was chosen as a 'worst-case' scenario making it highly translatable to clinical practice[43]. Sheep masticate for 7–8 h per day and unlike humans they cannot be tube fed or given a puree diet, thus there is accelerated mechanical stress applied to the scaffold that commences immediately after implantation. For these reasons, an unreconstructed

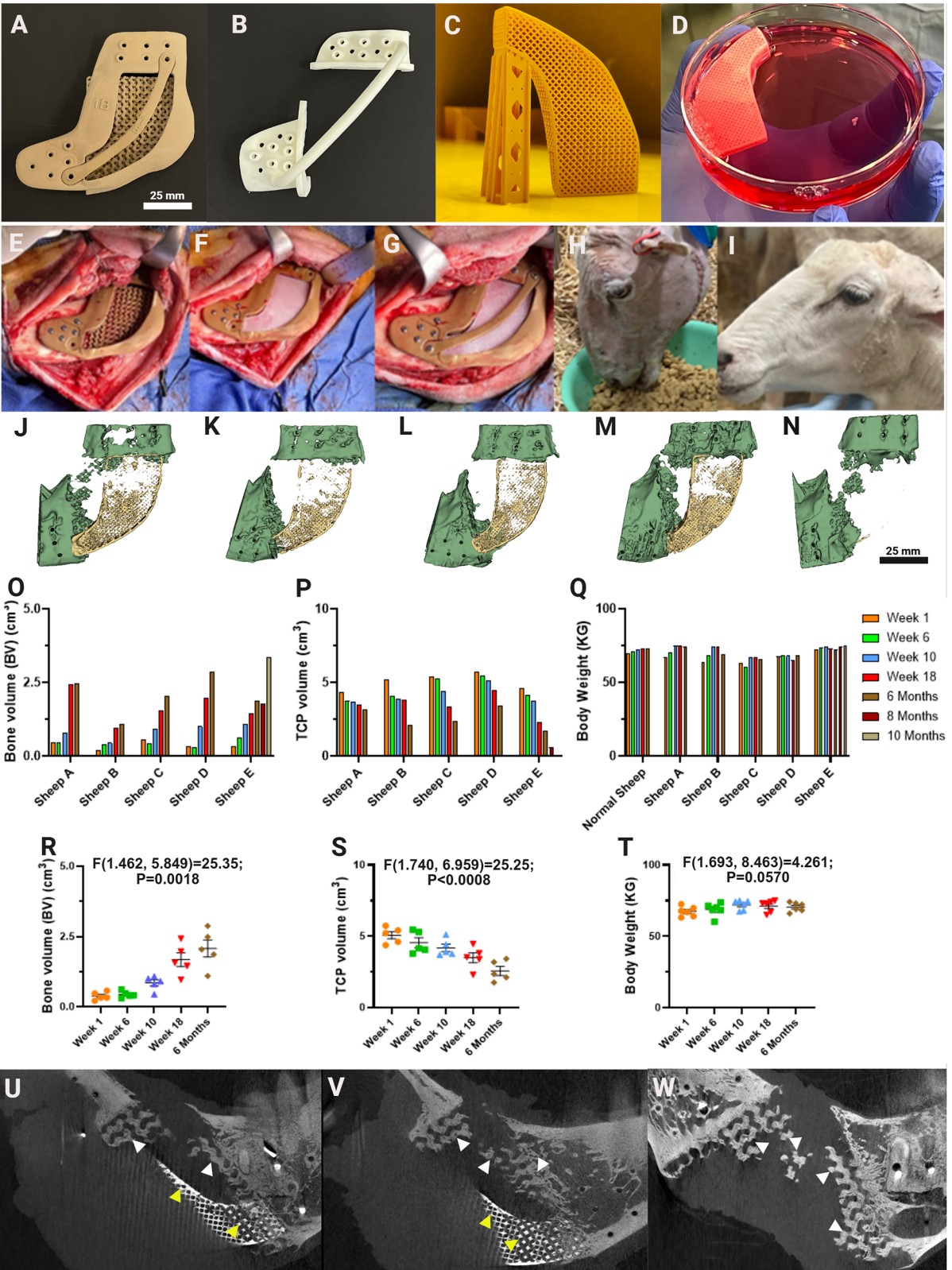

control would not be humane, however, weight-gain in implanted sheep matched that of normal unoperated sheep. Furthermore, mature sheep have variable regenerative capacity, better representing the spectrum of high-risk human applications. The posterior region of the mandible was selected because it has the highest stress during molar mastication; even greater than humans[43].

The surface properties of load-bearing implants play a crucial role in shaping the biological responses of living bone, influencing osseointegration[49]. More specifically, implant surfaces have been shown to regulate protein adsorption, platelet adhesion, and inflammatory responses, which subsequently affect osteogenic processes and bone remodelling[49]. Modifying surface roughness, chemistry, and

**Fig. 5 | Artificial bone assembly and surgical implantation with durable clinical performance and radiological evidence of in vivo osteogenesis within the gyroid structure. A–D** Components of the construct: PIII-treated LS-PEK frame (**A**); 3D-printed nylon-12 surgical guide for osteotomy and screw placement (**B**); βTCP lattice and supporting structure (**C**); and βTCP lattice with ADSC-laden GelMA in culture media (**D**). **E–G** Surgical implantation: LS-PEK frame inserted into a segmental mandibular defect (**E**), followed by βTCP lattice (**F**) and LS-PEK restraining crossbar (**G**). **H–I** Postoperative recovery: sheep begin oral intake immediately (**H**). All animals showed excellent tissue healing at 6 months, with no inflammation or implant failure (**I**). **J–N** 3D reconstructed CBCT images at 6 months (**J–M**) and 10 months (**N**) show progressive bone formation. Black arrows indicate new bone within LS-PEK; red arrows indicate new bone within the βTCP lattice. **O–P** Quantification: new bone volume increased significantly across all sheep (**A–E**)

over the implantation period, while βTCP lattice volume decreased, indicating concurrent bone ingrowth and material resorption. **Q** All sheep maintained or gained weight during implantation, similar to non-implanted controls. **R–T** Repeated-measures one-way ANOVA confirmed significant bone volume increases and βTCP resorption over time, with no significant change in body weight ($n = 5$). Grouped analysis: bone volume, $F(1.462, 5.849) = 25.35$, $p = 0.0018$; βTCP volume, $F(1.740, 6.959) = 25.25$, $p < 0.0008$; body weight, $F(1.693, 8.463) = 4.261$, $p = 0.0570$. Data are mean ± SEM. **U–W** Sagittal CBCT images at 6 months demonstrate new bone interlocking within LS-PEK gyroid spaces, consistent with device osseointegration and durable clinical performance. White arrows indicate bone within LS-PEK; yellow arrows indicate bone within βTCP lattice. Source data are provided as a Source Data file.

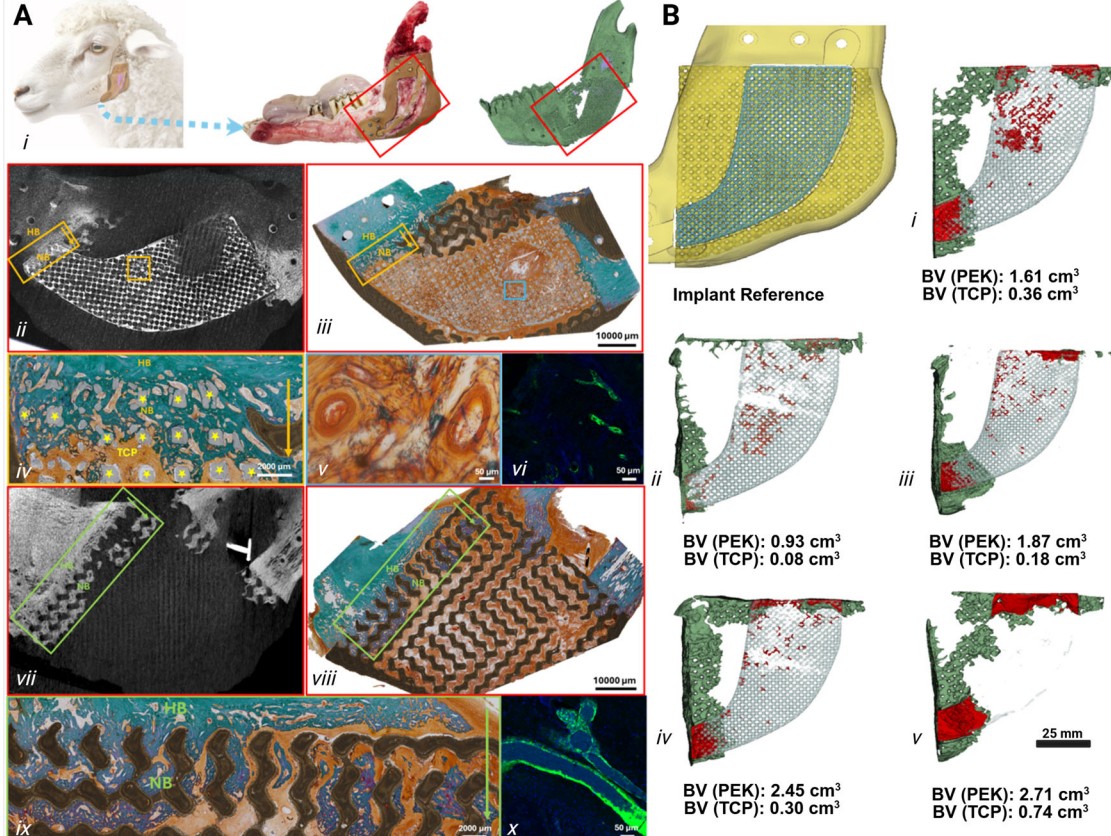

**Fig. 6 | Explant analysis demonstrating histological bone-implant contact and greater osteoconduction through the LS-PEK frame than the βTCP lattice. A** i. Location of the artificial bone implanted into the sheep mandible, macroscopic appearance of explanted mandible and artificial bone, and 3D reconstructed μCT image of explanted mandible with artificial bone. ii-ix. Resin embedded histomorphometry confirming μCT findings: New bone within βTCP lattice at anterior native bone-lattice interface (yellow rectangle) and within the middle of the lattice shown at low and high power on μCT (ii) and histology (iii, iv) (Sample size $n = 5$). HB refers to host bone and NB refers to new bone. The mature new bone near the interface shows a dense matrix. Neovascularisation within βTCP lattice on low power (iii: blue square) and high power (v). Vasculature was found at the interface between host bone and βTCP as shown by immunofluorescence for vWF (vi). New bone formation within the gyroid voids (green rectangle) of LS-PEK on μCT (vii) and histology at low

magnification (viii) and high magnification (ix). Vasculature was identified at the interface between host bone and the LS-PEK scaffold as shown by immunofluorescence for vWF (x). Scale bar: iii & viii- 1000 μm; v, vi & x- 50 μm; iv & ix- 2000 μm. **B** 3D reconstructed μCT images of explants performed at 6 months of implantation in sheep A-D (i - iv) and 12 months implantation in sheep E (v) demonstrating variable degrees of new bone formation despite reproducible clinical outcomes indicating that device performance was more dependent on osseointegration than amount of osteoconduction. In these images (i-v), green represents new bone within the LS-PEK frame, while red represents new bone within the βTCP lattice. Most of the new bone formed within the LS-PEK frame representing 11% to 38% of the available scaffold volume, and the greatest amount was observed in Sheep D (iv) and E (v). Created in BioRender. Crook, J. (https://BioRender.com/68by56e). Source data are provided as a Source Data file.

porosity have been shown to enhance implant performance by promoting osteoblast adhesion and differentiation. Micron- and submicron-level modifications improve cell attachment, while nanoscale features help regulate the biological microenvironment, influencing osteoblast activity and bone mineral deposition[50,51]. Furthermore, PIII-induced hydrophilicity promotes faster healing by

improving protein adsorption and cell migration, leading to better implant integration[52]. Whilst we were unable to include suitable controls for ethical reasons, based on our prior work the favourable outcomes could be principally attributed to stable osseointegration through applying LS technology to PEK, employing a non-abrasive blasting method to maintain the LS-PEK implant surface micro-

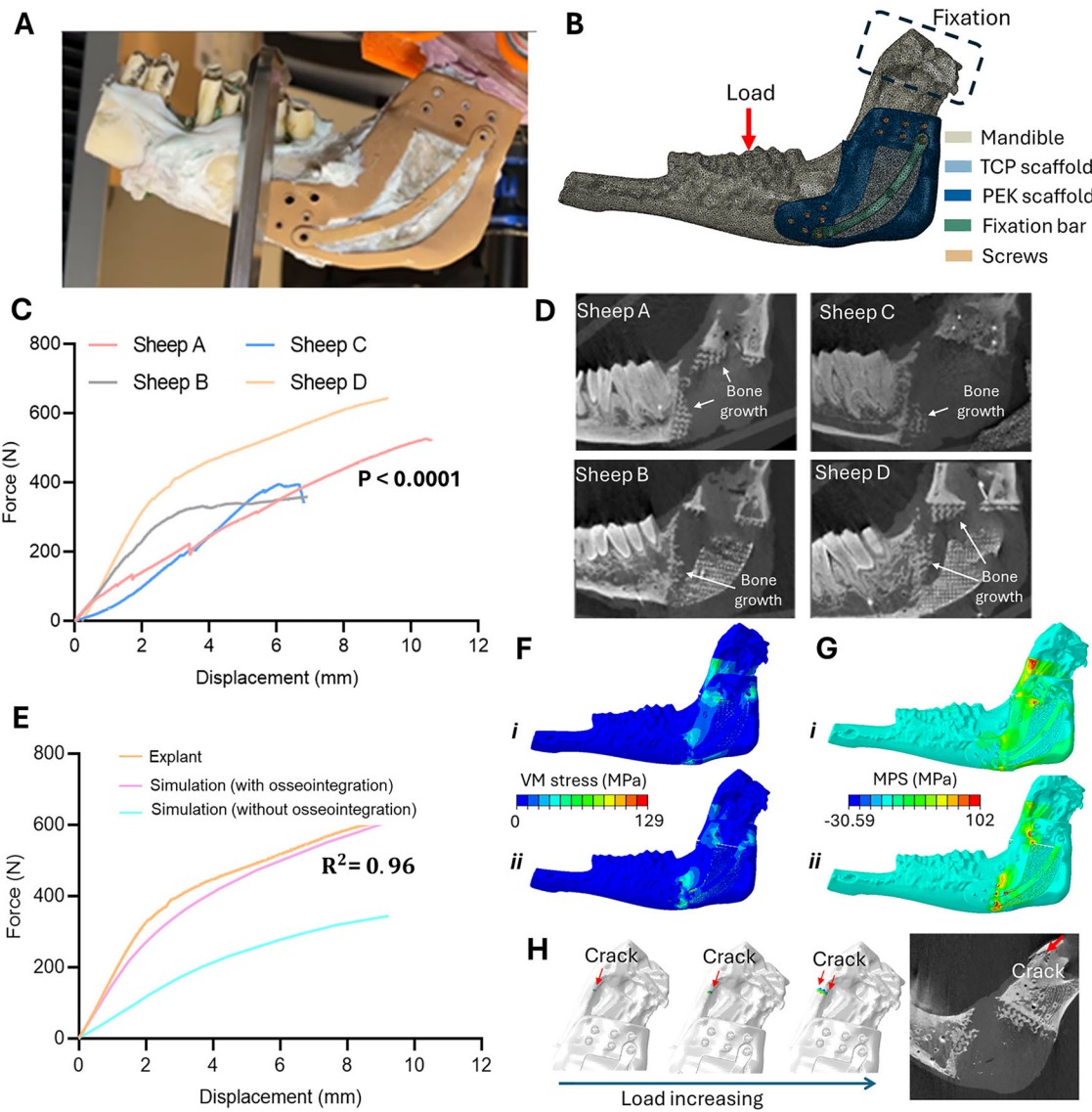

**Fig. 7 | Ex vivo mechanical testing and corresponding μCT demonstrating that mechanical performance matches FE modelling. A** Explant mechanical testing of the hemimandible with soft tissue and fixation screws removed. The coronoid process and condyle were fixed in resin and loading force applied to the molar teeth. **B** Representative FE model with loading and boundary conditions for the numerical simulation. **C** Force-displacement curves ($n = 4$) for the explanted sheep hemimandibles were compared using a non-linear (quadratic) least squares regression fit (F (9,101162) = 214333, $P < 0.0001$). Mechanical tests were stopped at the first indication of implant or bone failure to allow histological analysis of an intact specimen. Sheep D (643 N) and Sheep A (526 N) had the highest force at failure. In these samples the native mandible fractured before failure in the artificial bone or bone implant interface. The lowest force at failure was observed in Sheep B (395 N) and Sheep C (359 N). **D** μCT images for the corresponding sheep showing that greater new bone volume was associated with the higher fracture load values.

**E** Representative simulated force-displacement curve for the selected sheep model (Sheep D) with and without osseointegration (R2 = 0.96). Without osseointegration, the overall reconstructed system has lower stiffness (**F**). Distribution of VM equivalent stress in the sheep mandible-scaffold system with osseointegration (Fi) and without osseointegration (Fii). **G** Distribution of MPS in the sheep mandible-scaffold system with osseointegration (Gi) and without osseointegration (Gii). Higher MPS values indicate an increased risk of brittle fracture. **H** Simulated fracture path in the sheep mandible using XFEM with the corresponding μCT scan showing that the predicted fracture path obtained from XFEM matched that observed during mechanical testing as confirmed on μCT analysis. The crack originated in the posterior mandibular segment and extended towards the upper surface. The location of the crack aligns well with the areas of higher MPS, suggesting that high tensile stress could lead to brittle fracture. Source data are provided as a Source Data file.

roughness, and plasma (PIII) treatment to increase hydrophilicity and enhance bioactivity. Together, these properties, by design, improved cell adhesion and promoted strong osseointegration, resulting in long-term stability.

Despite in vitro mechanical stability, without osseointegration, implant failure will occur due to osteolytic screw loosening in response to unfavourable stress distributions at fixation points, thus demonstrating long term in vivo biomechanical stability was essential. Osseointegration is more important than osteoconduction in permanent scaffolds, since unlike bioresorbable scaffolds, the mechanical

properties do not degrade unless due to physical trauma from a forceful impact or ductile stress injury. So, the initial integration with the surrounding host bone is crucial for the implant to maintain mechanical anchorage. Here, ex-vivo mechanical testing showed that the bone-implant interface can withstand even greater tensile force than native bone. This is supported by histological and radiological evidence of new bone interlocking with the implant's gyroid structure. Time-dependent mechanobiological simulations suggest that osteoconduction was stress-driven due to the bone-compatible elastic modulus of LS-PEK. Although complete bone-bridging of the LS-PEK

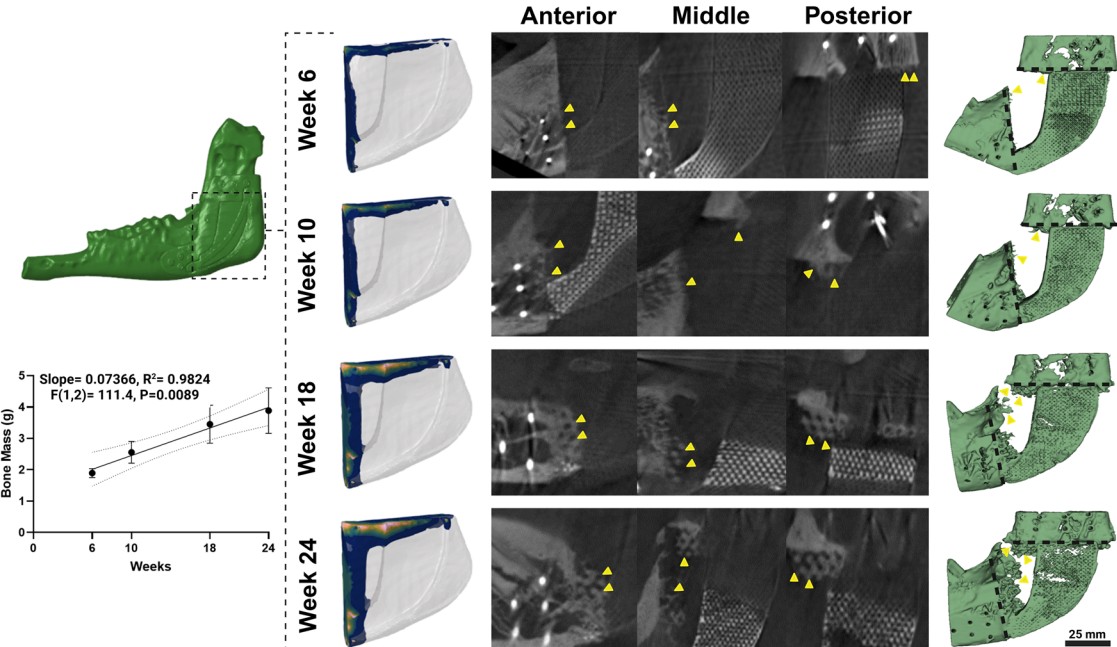

**Fig. 8 | Time-dependent mechanobiological numerical simulations match observed bone growth on CBCT.** Numerical simulations of bone growth within the scaffold assuming no mechanical contribution of the βTCP lattice (left; refer to Fig. 4E) best matched the in vivo osteogenesis observed in sheep D over a period of 6 months demonstrated on sagittal CBCT slices shown in the middle, and 3D reconstructed CBCT images on the right. At week 6, new bone formation began at the interface between the mature bone and the artificial bone anteriorly and posteriorly. By week 10, the newly formed bone had extended into the internal gyroid space of the anterior aspect of the LS-PEK frame (indicated by yellow arrows on the CBCT sagittal & 3D images, black dashed lines on the 3D images indicate the osteotomy plan) and the density of the newly formed bone around the interface area increased. By week 18, the newly formed bone had also extended convincingly into the gyroid space of the posterior aspect of the LS-PEK frame. By week 24, the

newly formed bone had bridged the full length of the LS-PEK frame in the regions of peak VM stress with the highest density of the newly formed bone being 0.8 g/cm³ and the lowest being 0.2 g/cm³. The predicted bone mass at Week 6, 10, 18 and 24 at shown, peaking at 3.062 g. Notably, the pattern of observed bone formation follows a similar trend as shown in Supplementary Movie 3. In this figure, dark blue indicates low-density bone (0.3 g/cm³), transitioning through green and yellow (0.5 g/cm³), to pink, which represents high-density bone (0.8 g/cm³). The corresponding colour bar is presented in Fig. 4 (E-G). Linear regression analysis established a significant relationship between bone growth and time after implantation, $F_{(1,2)} = 111.4$, $P = 0.0089$, with a slope of 0.07366 and R squared = 0.9824. Linear regression plot represents line of best fit (solid line) and 95% confidence interval (dotted line). Source data are provided as a Source Data file.

frame was not essential for successful reconstruction, lower new bone volume was associated with lower ex-vivo fracture load values, demonstrating the benefit of progressive osteoconduction in addition to osseointegration. FE simulations showed that osteogenesis also promotes load-sharing with native bone, preventing stress concentrations around screw holes that in clinical practice cause osteolysis and screw loosening. A significant mismatch in elastic modulus between the implant material and bone can cause stress shielding, where the implant bears most of the load, reducing mechanical stimulation to the surrounding bone, which decreases osteogenesis and increases the risk of implant failure[53]. Importantly, numerical modelling applying different loading and boundary conditions demonstrated variable osteogenic responses, matching the wide range of outcomes observed. This mechanically and biologically driven variation represents a challenge for bioresorbable scaffolds that depend on predictable rates of degradation and bone ingrowth. These results support our hypothesis that a fracture-tough bone-like structure that osseointegrates and promotes stress-induced osteoconduction, such as thermally toughened PIII-treated LS-PEK, is a better strategy than bioresorbable scaffolds with rigid plates due to their variable degradation rates, particularly in high-tensile applications and in oncology where metal plate fixation is undesirable.

The variable rates of new bone formation and βTCP resorption observed in this study affirm the need for achieving optimal degradation kinetics for effective tissue regeneration and stability of the bone-scaffold system[54,55]. Whilst various bony defects have been reconstructed with bioresorbable 3D-printed polycaprolactone and

βTCP fixated with Ti plates[21,53], there are no studies of 3D-printed βTCP scaffolds being implanted into the sheep mandible and hence the degradation rate of βTCP fabricated using this approach was previously unknown. βTCP is insoluble at physiological pH, thus, its bioavailability is dependent on the presence of active osteoclasts and other factors such as steam sterilization[56]. We opted to dry-heat sterilize to prevent conformational change of the lattice, which required a precise fit within the LS-PEK frame with minimal tolerance. Thus, the osteogenically differentiated ADSCs were infused to enhance the bioavailability of the calcium reservoir and host bone ingrowth may have been augmented by the release of calcium (and phosphate) ions during βTCP scaffold degradation. μCT-based bone volume estimates may be unreliable because the βTCP lattice is radio-opaque and degradation products may be misinterpreted as new bone using μCT. It is challenging to separate the thresholding limits for calcium which is present in both βTCP and in newly formed bone. To overcome this, we combined μCT with histology to provide more reliable quantification. The presence of new bone and neovascularization within the βTCP lattice could be identified histologically but we were unable to determine whether osteoblasts originated from the host or the ADSCs. Regardless, the volume of new bone generated within the βTCP lattice was insufficient for mechanical stability and the rate of βTCP degradation was highly variable. This confirmed our concern that bioresorbable scaffolds will always require permanent structural support unless the scaffold materials degrade at a rate that matches new bone formation during the healing process[57]. If the degradation is too rapid, the scaffold's porous structure may collapse, hindering mass transfer.

Conversely, if the degradation is too slow, it can result in fibrous capsule formation and poor integration with the host tissue.

Further work is required to evaluate this technology for its intended purpose in the oncological setting. Osteoinduction could be enhanced through the addition of bone morphogenetic protein 2 (BMP-2), but growth hormones have the potential to accelerate cancer growth, reinforcing the value of stress-driven osteogenesis as a safe and clinically viable approach. Radiotherapy is commonly used in the treatment of cancer, but the effect of radiotherapy will be difficult to test without compromising animal welfare. Despite this, radiolucent PEK avoids the challenges of local dose 'hot spots' during radiotherapy caused by high atomic number materials such as Ti, which may lead to tissue necrosis[58]. Soft tissue integration is equally important for long-term success. We observed strong tissue adhesion to the artificial bone but optimizing the surface topology for soft tissue integration is needed to facilitate more advanced reconstructive goals, such as dental rehabilitation. In this regard, sufficient new bone remote from the bone-implant interface is also needed to integrate dental implants that support dental prosthetics.

In conclusion, patient-matched SGBR using 3D-printed artificial bone with permanent structural integrity, enhanced osseointegration, and stress-driven osteoconduction is a translatable paradigm in SGBR for segmental bone defect repair. This strategy may be more reliable than bioresorbable SGBR because it is less dependent on synchronized rates of osteogenesis and scaffold degradation, both of which are highly variable within and between individuals. However, the concept of combining bone-like permanent scaffolds, such as LS-PEK, with osteoinductive bioresorbable scaffolds to accelerate osteogenesis is worth exploring further because there are clear biomechanical advantages to achieving higher bone volumes within the scaffold.

## Methods

### Ethical statement and preclinical model

All the animal experiments involved in this work were approved by the University of Sydney Animal Ethics Committee (approval no. 20221212). Six mature female sheep (Ovis Aries) aged 7–8 years were selected due to the anatomical, histological, physiological (bone turnover), and biomechanical similarities with the human mandible[59–62]. A minimum implantation period of 6 months was chosen because either clinical, radiological, or histological evidence of implant failure would become evident during this period[63]. One sheep's implantation period was extended to 12 months to see if additional new bone would form given more time. For ethical reasons, a control group with unreconstructed segmental defects could not be included in the study.

Mature sheep maintain a stable weight and display manageable handling characteristics. However, ruminants continuously chew 7–8 h per day with a cyclical motion, which places additional mechanical demands on the jaw and implants despite similar bite-forces to humans. A critical sized 6 cm defect was selected because it is a common length in oncological jaw reconstruction. The angle of the mandible was selected because it is a site of maximum stress and strain during intercuspal (bilateral) and unilateral clenching and has similar anatomical features to humans, particularly the ratio of cortical to cancellous bone[43,64]. This also allowed implantation outside of the tooth-bearing region to avoid interfering with nutrition and contamination by oral bacteria.

### Sawtooth cylinder, non-segmental mandibular scaffold, and in vivo bioreactor design

The design and development of implants used in Fig. 2 employed 3ds Max 2020 (Autodesk, Inc., San Francisco, CA, USA) utilizing the polygonal modelling technique. The sawtooth implant was designed with a 5 mm diameter and height cylinder, featuring nine grooves to create a saw-tooth structure, which increases surface area and supports both single and double fixation sites. The digital models were generated by mirroring the left-side models, ensuring symmetry and anatomical accuracy. Sawtooth cylinders were composed of PIII-treated LS-PEK and PBF-LB grade 23 Ti. For non-segmental mandibular scaffold designs, LS and FFF techniques were used for PEK and PEEK, respectively. A 3D CAD model of the mandible was generated from CT scan data of a 6-year-old female Dorset-cross sheep. Segmentation of the cortical bone, trabecular bone, and teeth was done using ScanIP™ software (Synopsys, USA). A partial defect was created along the inferior border of the ramus using Boolean subtraction operations, and the CAD model was transferred to SolidWorks™ (Dassault Systèmes, France) to finalize the design. The non-segmental mandibular scaffold implant is a one-body structure designed to fill the defect with a scaffold, while external wings (flanges) secure the implant to the bone surface using screws. The scaffold's structure is a 3D array of periodic unit cells based on Schwarz P-surfaces with a pore size of 1 mm and strut size of 1 mm. The resulting porosity was 50% as determined by volume fraction. Schwarz P is a class of TPMS optimised for tissue integration and vascularization. In the scapular studies, double-layered bioreactors were used. These 10 × 10 × 5 mm bioreactors were designed to enable contact with both the bone and periosteum in the scapula. Each scapula accommodated four rows of six bioreactors, allowing for a total of 96 bioreactors per sheep, providing an adequately powered model for in vivo evaluation of biomaterial performance in bone regeneration.

### Artificial bone design

The artificial bone has three components: A PIII-treated LS-PEK frame, an internal βTCP lattice infilled with ADSC-laden GelMA, and a LS-PEK crossbar used to secure the βTCP lattice (Supplementary Fig. 7). The LS-PEK frame was custom designed to fit each sheep's left mandible using a clinically validated VSP process where the ablation and reconstruction are digitally simulated[5]. Prototypes were designed based on helical CT scans (1 mm slice thickness) of cadaveric sheep mandibles and the final artificial bone design was based on high-resolution C-arm CBCT (Siemens Artis Pheno, Siemens Healthcare GmbH, Erlangen, Germany) acquired from live sheep adhering to an industry-standard protocol[65]. 3D stereolithographic models (.stl) of the mandible and teeth were created from the Digital Imaging and Communications in Medicine (DICOM) imaging files using Materialise Mimics 24 (Materialise NV, Leuven, Belgium). The LS-PEK frame model was created using polygonal modelling techniques in 3ds Max 2022 (Autodesk, Mill Valley, California, U.S.). After analysing the anatomical models, planned osteotomy planes were defined to virtually cut the sheep mandible, ensuring an ideal matching interface between the implant and the bone. Anterior and posterior solid flanges were created to conform to the mandible surface curvatures with a 3.2 mm thickness. Six reinforced 2.2 mm diameter screw holes were defined and cut using Boolean subtraction in each flange. The middle section bridges the bone defect employing a single gyroid TPMS scaffold architecture (3.5 mm unit size and 1.6 mm pore size) with solid reinforcement superiorly and inferiorly. The strut size is defined as the minimum diameter/thickness of the cross section of a strut while the pore size was defined as the maximum diameter sphere that could fit through the pores of the lattice. A 50% volume fraction porosity level was chosen to give equal weighting to the strength of the struts and the cleanability of the residual powder after printing. A 3.5 mm cell size was chosen after a series of test lattices of varying sizing from 5 mm cell sizes down to 2 mm cell sizes were designed, printed, and cleaned. It was observed that cell sizes below 3.5 mm were difficult to clean effectively and cell sizes below 3 mm had struts break during the cleaning process. Gyroid replaced the Schwartz P structure used in phase 2 studies due to its isotropic mechanical properties and better termination at the surface of the implant. Gyroid structures are also

stronger[66] and promote better osteoconduction[49]. A chamber to accommodate the βTCP lattice was designed with a 1 mm gap for installation.

The βTCP lattice formed the core of the artificial bone and traversed the LS-PEK frame to contact the cut bone at each end held in place with a 2 × 4 mm thick LS-PEK crossbar which engaged one screw hole in each flange (Supplementary Fig. S7). The cell size of the Body-Centered Cubic βTCP lattice was selected to be 1.875 mm, with a beam thickness of 0.4 mm; these specifications were within the limits of the 3D printer used (Lithoz Cerafab 7500), allowed for effective post-processing (removal of residual ceramic slurry), provided adequate structural integrity, and permitted the lattice space to be backfilled with GelMA hydrogel. The design process used nTopology, a specialised lattice generation software, to create a lattice structure within the confines of the desired geometry to ensure that the lattice would align with the boundaries of the planned insertion void. Support structures are required to provide a foundation for subsequent material layers during the printing process, however, unsintered parts are delicate, and the removal of support material risks damage. Numerous lattice geometries were evaluated, leading to the selection of a Body-Centered Cubic lattice, rotated 45 degrees relative to the planned printing direction to minimize unsupported geometries and eliminate the need for internal support structures. To avoid lattice beams that are not connected to terminating geometries at organically shaped boundaries, a warping operation was performed to stretch close-proximity unit cells to the boundary face, causing the lattice units to morph to the outer boundary, eliminating partially cut unit cells. Along acute edges, this warping function can cause undesirable densities as numerous unit cells are pulled to that edge. To mitigate this, a fine border was placed along these acute angles.

### Image-based FE model
The 3D computational model of the sheep mandible was created based upon the CT images by using the commercial code ScanIP (Synopsys Simpleware, Mountain View, CA). After surface smoothing and refinement, the 3D solid models of mandible, scaffold, and screws were meshed using 4-node tetrahedral elements in ScanIP, containing 3,826,530 elements with a total of 2,092,518 degrees of freedom (DOFs). The commercial FE code Abaqus (2016, ABAQUS Inc, Providence, RI) was adopted to conduct the subsequent FE analysis[38,45]. All materials were assumed to be isotropic and linear elastic. The average Young's modulus of mandibular bone ($E_{bone}$) was assumed to be 12,000 $MPa$ and the difference in the predicted biomechanics of teeth and cortical bone was ignored[45]. The Young's modulus given to LS-PEK ($E_{PEK}$) and ($E_{TCP}$) was 2,400 $MPa$, and 15,000 $\bar{v}$, respectively. A Poisson's ratio ($\bar{v}$) of 0.3 was used for all bone tissues and the scaffold.

### Loading and boundary conditions
Mechanical laboratory testing was conducted to validate the FE model where the load and boundary conditions were prescribed to match those used in the actual mechanical test. The sheep condylar region was immobilised (fixed in all degrees of freedom) and a downforce was applied on the premolar teeth area. Two osseointegration conditions were considered: the first scenario assumed a lack of osseointegration, while the second scenario mimicked the presence of osseointegration between the scaffold and the host mandible. Therefore, in the first scenario, a frictional contact with a coefficient of 0.1 was applied at the bone-scaffold region. Conversely, in the second scenario, a tie constraint was applied at the bone-scaffold interface to represent the state of osseointegration. A friction grip connection with a coefficient of 0.3 was applied at the interface between the bone and implant sheath when contact occurred. The resultant vertical displacement was extracted from the FE simulation and used to calculate the stiffness of the sheep mandible-scaffold system.

### XFEM modelling
The eXtended Finite Element Method (XFEM) was used to simulate the crack initiation and further propagation[42]. A key advantage of XFEM for fracture studies is its independence of the element mesh from the morphology of the crack surface and crack front[44,67]. In the XFEM simulation, the MPS criterion was selected as the crack initiation criterion[68], represented in the following equation:

$$f^e = \frac{\sigma_1^e}{\sigma_{max}^0} \tag{1}$$

where $\sigma_{max}^0$ refers to the maximum allowable principal stress (tensile stress), $\sigma_1^e$ is the MPS in element ($e$), and $f^e$ is the stress ratio, which determines if cracking will initiate in the element. A crack was assumed to occur when the MPS exceeded the pre-defined tensile strength of the material, 150 $MPa$ and 80 $MPa$ for bone and LS-PEK, respectively. The subsequent crack growth was associated with the fracture toughness, 1.46 $MPa·m^{0.5}$ and 1.05 $MPa·m^{0.5}$ for bone and LS-PEK respectively.

### Mechanobiological model for tissue growth
To simulate bone growth into the scaffolds, a time-dependent dynamic mechanobiological model was developed. In this model, the porous scaffold region was treated as a solid and homogenized medium at the macroscopic level (Supplementary Fig. 8). The mechanobiological model for tissue growth operates at two scales: the microscopic level and the macroscopic level. The load and boundary conditions were set to simulate the biting forces for the sheep mandible[43].

At the microscopic level, the lattices of LS-PEK and βTCP scaffolds were meshed into cubic-based unit cells. FE-based homogenization analysis was then conducted on these unit cells. It is important to note that the newly formed bone within a unit cell can significantly affect the structure of these cells. Therefore, a uniform sampling approach[69] was employed to calculate the homogenized constitutive matrix and diffusivity matrix, considering bone densities ranging from 0.05 g/cm³ to 1.62 g/cm³. At the macroscopic level, the effective stress, calculated using the Cauchy stress and the homogenized constitutive matrix of each unit cell, was used as the stimulus for driving bone tissue growth within the scaffold[45]. The bone deposition rate within the scaffold was calculated as follows,

$$\dot{\varphi}_r = \begin{cases} c_s \lambda_r (\psi_r^{(n)} - \bar{\psi} - l), & \text{if } \psi_r^{(n)} - \bar{\psi} > l \\ 0, & \text{otherwise} \end{cases} \tag{2}$$

where $c_s$ = 0.6 is an empirical constant[45], $\lambda_r$ denotes the normalized concentration of osteoblasts with a maximum concentration of 100%, which is calculated by solving Fick's law, using the homogenized diffusivity tensor, $\bar{\psi}$ is a reference stimulus level, and $l$ is the lazy zone around the reference stimulus level, and $\psi_r^{(n)}$ is the mechanical stimulus calculated from the effective stress during loading circles[70–72].

### Surgical guide design and 3D printing
A bridged nylon-12 surgical guide was created using 3dsMax 2022 Software based on the LS-PEK frame design (Fig. 5B and Supplementary Fig. 7). Drill holes on the surgical guide were created based on the screw positions with irrigation channels. Sheep-specific surgical guides were printed in Nylon 12 (PA2200) using the FORMIGA P 110 Velocis Printer (EOS, Krailing, Germany) and heat-sterilised.

### 3D printing of FFF-PEEK and LS-PEK implants and heat toughening of LS-PEK frames
PEEK, PEKK, and PEK are three distinct members of the PAEK family. The chemical structures of PEEK and PEK are shown in Supplementary Fig. 9. PEEK implants were designed using firmware v3.3.5 and fabricated via FFF on an AON-M.2 3D printer from AON3D, Montreal,

Canada with 1.75 mm Thermax PEEK (batch 49-080620-06JV) filament produced by 3DXTech, Grand Rapids, Michigan. PEK (commercially labelled as PEEK HP3) powder (EOS GmbH, Krailling, Germany) was LS-printed using an EOS GmbH P800 (Krailling, Germany) 3D printer system. The EOS P800 has a maximum build volume of 700 × 380 × 560 mm. Accounting for 5% shrinkage, this allows for a maximum scaffold size of 666 × 362 × 533 mm, more than enough for most bone-based scaffold applications. Meanwhile, the minimum scaffold size is dependent on the design. The minimum feature size the P800 can produce is ~0.8 mm which is appropriate for all bones except those of the middle ear. The powder was stored in 45% humidity at 22 °C until printing. The building parameters of the LS-PEK frames are listed in Supplementary Table 1. The laser parameters used to sinter the parts are custom and the result of prior mechanical testing studies. After printing was complete the cake was left to cool with the default high temperature cooldown routine until the temperatures fell below 60 °C. LS-PEK frames were removed from the powder cake and blasted free of loosely sintered powder with shaved solid CO2 (Dry Ice) using a ColdJet MicroClean (ColdJet, Moorebank, Australia) at 1.8 bar gauge pressure and a feed rate of 0.15 kg/min. LS-PEK frames were thermally toughened using a bespoke quenching furnace preheated to a set temperature, called the heat soak temperature (Supplementary Fig. 10). After the heat soak time elapsed, the frames were rapidly quenched by blasting with cool dry air. The system parameters for the quenching process are provided in Supplementary Table 2.

## PIII pre-treatment, PIII treatment, ultrasonication cleaning, and sterilization of LS-PEK frames

PIII was used to reduce the electrostatic attractive force between loose powder particles and the implants to increase cleaning efficiency ("pre-treatment") prior to ultrasonication, and to enhance biocompatibility and osseointegration of the artificial bone ("main PIII treatment")[25–27]. PIII treatment was administered via dielectric barrier discharge[73]. The LS-PEK frames were suspended in a large bell jar (borosilicate, 24 L volume) either centrally for the pre-treatment, or as close to the glass wall and exterior electrode as possible for the main PIII treatment. The chamber was evacuated to at least $2.0 \times 10^{-4}$ mbar, then the specimens were submerged in pure nitrogen gas at various pressures. A Rup-6 power supply (GBS Elektronik, Radeberg, Germany) was used to apply negative pulses of 10 kV amplitude, a rise time of 300 ns, 20 µs pulse length and a pulse repetition frequency of 1000 Hz to excite the discharge. The pulses were applied to an exterior aluminum foil electrode covering ~50% of the glass bell jar. The regime for the pre-treatment consisted of 4 min at 0.437 mbar and 4 min at 0.937 mbar nitrogen gas pressure. The regime for the main PIII treatment was 5 min at 0.437 mbar, 5 min at 0.250 mbar, 5 min at 0.900 mbar and 5 min at 0.400 mbar. The rationale for the variations in the pressure of the process was to ensure adequate plasma penetration of the interior porosities so that PIII treatment is applied to all surfaces, both interior and exterior. Adequate treatment of the interior surfaces of the gyroid structure is especially important for osseointegration and osteoconduction. To further improve the uniformity of the PIII surface treatment, the LS-PEK frames were turned over to present a new orientation to the plasma, the chamber was re-evacuated, and the treatment procedure was repeated for a total treatment time of 40 min. After PIII pre-treatment the LS-PEK frames were ultrasonicated in an Ultrasonic Cleaner (180 W, 40 kHz; Vevor, Shanghai, China) in sterile water for 30 mins at 40 °C and cleared with compressed air. Then, the implants were air dried overnight before the main PIII treatment. After the main PIII treatment, the scaffolds were steam sterilized by an autoclave at 134 °C for 5 min at an absolute pressure of 3077 mbar in sealed sterilization pouches.

## 3D printing of βTCP lattices

The manufacturing of βTCP lattices was performed using the Lithoz Cerafab 7500 3D printer at the ANFF Materials Node, located within the University of Wollongong's Innovation Campus. This machine employs a light projector to selectively polymerize layers of a photopolymer onto a platform. It utilizes a ceramic-laden 'slurry' and a high-precision UV light source to successively build layers until the part is fully formed. After printing, the parts were transferred to a washing station, which is an enclosed cabinet equipped with a vapor extraction system and a precision air brush. The air brush was used by the operator to dispense a proprietary cleaning solution onto the parts, aiding in the removal of the viscous residual material. If not adequately removed, the residual uncured ceramic slurry material will sinter to the part during thermal post-processing, potentially causing geometrical abnormalities and blocked lattice cells. Thermal post-processing was conducted after the parts have been cleaned of residual slurry. This process involves two furnaces. The first furnace, a high-precision unit, carefully elevates the temperature of the parts to remove volatile components of the polymer matrix. The second furnace, a higher temperature unit, solidifies the remaining ceramic material through crystallization. The lattices are physically complete and sterile once the sintering process is finished. To minimize the risk of contamination, parts were transferred using gloves from the furnace directly into sealable bags.

## Harvesting, expansion, and encapsulation of ADSCs in GelMA

~50 g of adipose tissue was harvested from the back of anesthetized sheep using a scalpel, placed in phosphate-buffered saline (PBS) containing 5% penicillin/streptomycin (P/S; Thermofisher Scientific) and maintained cold (4 °C) on wet ice during transport. Under sterile conditions in a Biological Safety Cabinet, the collected tissue was washed several times with PBS containing 5% P/S, drained and transferred to a sterile tissue culture dish and minced finely using a scalpel blade. The resultant minced tissue was weighed and transferred to a tube with equal (w/v) 0.075% collagenase IV (Sigma) in Dulbecco's modified Eagle medium (DMEM, Thermofisher Scientific) containing 2% P/S. The tissue was incubated for 2 hr at 37 °C with gentle agitation every 5 min and pipetting up and down several times every 30 min using a 25 ml serological pipette. After digestion, an equal volume of 20% fetal bovine serum (FBS; Thermofisher Scientific) and 2% P/S in DMEM were added to neutralize enzyme activity, followed by centrifugation at 760 x g for 10 min. The fat layer was removed from the tube and the collagenase solution aspirated, followed by resuspension of the pellet in an equal volume of DMEM containing 1%P/S. Samples were again centrifuged at 760 x g for 10 min and the supernatant aspirated without disturbing the cell pellet. The pellet was resuspended in DMEM with 10% FBS and 1% P/S and filtered using a 100 µm nylon cell strainer. Cells were counted and seeded at a minimum cell density of $10 \times 10^3$ cells per $cm^2$ in tissue culture flasks in DMEM with 10% FBS, 1% P/S and 10 ng/ml fibroblast growth factor 2 (bFGF; Thermofisher Scientific). Non-adherent cells were removed after 72–96 h culture, with medium changed every 3-4 days. Once cells reached 90% confluence, adherent cells were harvested by trypsinization (0.25% trypsin-EDTA, Thermofisher Scientific) before encapsulation within GelMA (TRICEP, Wollongong, Australia) at a final concentration of 5%, with 0.25% lithium phenyl-2,4,6-trimethylbenzoylphosphinate (LAP) and $10 \times 10^6$ ADSCs/mL. GelMA solution (derived from porcine skin gelatin, type A, 300 bloom, 37% degree of functionalization) was prepared by dissolving freeze dried GelMA solution, (prepared from porcine gelatine in PBS, mixed with methacrylic anhydride and reacted at 50 °C for 3 h, dialysed in distilled water at 40 °C for 1 week, and freeze dried) in PBS. The solution was mixed and heated to 37 °C for 1 h, and stored overnight at 4 °C, before returning to 37 °C for 1 h to fully dissolve[74].

### Infusion, culture, and differentiation of ADSC-laden GelMA into TCP scaffolds

β-TCP lattices were dry heat sterilized in sealed sterilization pouches (double bagged) at 170 °C for one hour, with a gradual ramp up and ramp down at 10 °C per 10 min to minimize warping of the lattice. ADSC-laden GelMA was infused into β-TCP lattices placed in a form-fitting 3D-printed resin mold using a syringe with an 18 G needle. Crosslinking was performed after infusing using a 405 nm photocuring toolhead of a CELLINK BIOX6 bioprinter (CELLINK, Gothenburg, Sweden). UV was delivered from a height of 1 cm above the scaffold for 120 sec applied at 0.75 cm intervals across the surface in a raster scanning fashion to treat the entire scaffold surface. Crosslinked constructs were then cultured in 10% FBS, 1% P/S and 10 ng/ml bFGF in DMEM for 7 days in 2 cm deep, 10 cm diameter tissue culture dishes at 37 °C in a 5% $CO_2$ humidified incubator for further cell expansion in situ. To pre-differentiate ADSCs in constructs to osteogenic lineage, culture media was replaced with osteogenic pre-differentiation media consisting of DMEM supplemented with 10% FBS (Thermofisher Scientific), 1% P/S (Thermofisher Scientific), 50 μM L-ascorbic acid 2-phosphate sesquimagnesium salt (Sigma), 10 mM β-glycerol phosphate (Sigma), and 100 nM dexamethasone (Sigma). ADSC-laden GelMA within TCP constructs were cultured at 37 °C in a 5% $CO_2$ humidified incubator for an additional 21 days, with a half medium change every 3-4 days, before transplantation to sheep and assembly within the LS-PEK frame.

### Surgical implantation

All sheep were anaesthetized, intubated orally, placed in a sternal recumbent position, and given intravenous antibiotics (cephazolin/1 g) at induction[38]. The sheep were prepped with sequential chlorhexidine (5%) and povidone iodine (7.5%) from neck to nose excluding the eyes and draped with exclusion of the endotracheal tube and the head was angulated downwards to allow saliva to run out of the field into a bucket. A 15 cm incision was made inferior to the lower border of the mandible and extended up posteriorly to follow the contour of the mandible. The facial vein was divided, and the marginal nerve reflected superiorly to expose the lower border of the mandible. The masseter and periosteum were stripped from the lateral aspect of the mandible superiorly to the sigmoid notch. The medial pterygoid was stripped on the medial aspect of the mandible to the lingula, where the inferior alveolar artery was identified and ligated. Sterile sheep-specific nylon surgical guides were applied to the lateral aspect of the mandible and secured with Ti screws (Supplementary Fig. 1). The guides were used to create 12 (six anterior and six posterior) full thickness screw holes using a 1.4 mm drill. A 6 cm segmental defect was created (Supplementary Fig. 1) using a reciprocating saw following the flanges of the surgical guide ensuring that there was no breach of the mucosa and protecting the lingual nerve. The surgical guide was removed and the sheep specific LS-PEK frame was removed from the sterile packaging and secured with 10 bicortical Ti screws (2 mm diameter) leaving two screw holes fallow. The βTCP lattice infilled with ADSC-laden GelMA was removed from culture media and installed into the cavity of the LS-PEK frame (Supplementary Fig. 1). The lattice was secured using the LS-PEK bar which was fixated using the two remaining screw holes (Supplementary Fig. 1). The pterygomasseteric sling was then reconstituted using 2/0 polyglactin sutures to cover the hybrid scaffold. The wound was closed in layers with 3/0 polyglactin and 3/0 glycolide and e-caprolactone copolymer sutures and dressed Opsite® spray. Following wound closure, a baseline CBCT was performed and then the sheep recovered.

### Recovery and diet

Following the implantation surgery, the anaesthesia was reversed according to the Laboratory Animal Services Anaesthetic and Post-operative Care Sheep standard operating procedure (SOP-OTH_11_Anaesthetic recovery and post-op care_sheep_20210527). The Laboratory Animal Services veterinary team carried out regular assessments on the following parameters: demeanour, vitals (temperature and heart rate), wound appearance (swelling, discharge, bleeding, dehiscence), tissue health (swelling, bleeding, formation of pus showing infection), hydration and regular pain assessments. The assessments were performed multiple times per day during the acute post-operative period; and the frequency was reduced as the wound healed and recovered. Postoperatively, the sheep received long-acting ceftiofur (10 mg/kg subcutaneously) to provide seven days antibiotic coverage. Multimodal analgesia was also administered by the veterinary team and a sheep pain scoring system was used regularly for sheep pain assessment and recorded on the Mandible Daily Hospitalization Monitoring Sheets. The sheep were given a standard diet consisting of a 50:50 mixture of lucerne and oaten hay at a rate of 2 kg per sheep per day. Additionally, they received ~3% of their body weight in oaten hay five days per week and lucerne hay two days per week and water was always available. Sheep had access to straw bedding, which they chewed and consumed throughout the day, and were given one to two handfuls of high-fiber sheep pellets and/or cracked lupin grain as treats. The weight of each animal was recorded monthly.

### CBCT scanning

A CBCT scan of the sheep's head, targeting the region of interest, was conducted using the Siemens ARTIS system (Siemens Healthineers AG). The scan had a duration of 16 seconds, operating at a peak kilovoltage of 61.3 kV and a tube current of 362.5 mA. The imaging data were saved in DICOM (Digital Imaging and Communications in Medicine) format for subsequent segmentation. CBCT imaging was performed on the day of implantation and then at weeks 6, 10, 18, and 24 for sheep A-D. For sheep E, two additional scans were carried out at 8 months and 10 months, providing extended follow-up data.

### Sheep euthanasia and scaffold harvesting

Upon conclusion of implantation period (six or 12 months), the sheep were sedated and euthanized according to the standard operation procedure for Euthanasia and Humane Killing of Pigs and Sheep from Laboratory Animal Services at Charles Perkin Centre, The University of Sydney (SOP-EUT_03_LAS Euthanasia and Humane Killing Sheep Pigs_20201015). CBCT scans were acquired and then the soft tissue overlying the implant was completely removed to inspect the artificial bone and tissue response to implantation. The fixation screws were removed, and a bone saw was used to cut the mandible between the canine and premolar and the hemi-mandible was removed by dividing the masticatory muscles and disarticulating the temporomandibular joint. Once harvested, the hemi-mandible was soaked in saline and stored at 4 °C before being delivered to μCT scanner and the mechanical testing laboratory.

### μCT scanning, image acquisition, and segmentation

μCT was conducted using MILabs' low-dose, ultra-high resolution X-RAY CT (MILabs B.V. - A Rigaku company, Duwboot 7a, 3991 CD Houten, Netherlands) at the preclinical facilities at Sydney Imaging, The University of Sydney. The sheep mandible with the implant was wrapped in Parafilm (Amcor, Thurgauerstrasse 34 CH-8050, Zürich, Switzerland) and placed in the medium-sized specimen bed connected to a movable docking station. The acquisition graphical user interface (GUI) displayed on a touch screen, where the scan volume was selected from optical camera views. The sample was scanned in Accurate mode with a 0.5-degree step angle, one projection per step, using 60 kV tube voltage and a 220 ms exposure time. The scan took 9 min and 44 seconds to complete. After scanning, μCT reconstruction was performed at 50-micrometer resolution using MILabs Reconstruction 13.12 software on a workstation running Windows Server 2008 R2 Enterprise (64-bit) with an Intel Xeon CPU E5-2690 @ 2.6 GHz and 128 GB of RAM. The result was then exported as a NIfTI

(Neuroimaging Informatics Technology Initiative) format file. Reconstructed NIfTI image files for each sheep mandibular sample were analyzed using 3D Slicer® image computing software (version 5.7.0-2024-06-18) to determine bone volume. Upon uploading each NIfTI file into 3D Slicer®, a region of interest (ROI) was defined using the "volume rendering" and "crop volume" options. The cropped image volume was then processed with a median image filter to reduce noise. Different materials at the defect site were distinguished by their respective threshold values. Given that LS-PEK is radiolucent, the primary task was to differentiate bone from the βTCP lattice. No fixed threshold could accurately delineate new bone from the high radio density βTCP without erroneously including volume near the surface of the βTCP due to partial volume averaging, beam hardening, and scatter effects. The bone threshold process was chosen and verified by aligning the μCT with histology images that confirmed the borders of bone, βTCP, and LS-PEK defect size. A CT value threshold range of 1800-5300 was selected to segment newly formed bone within the defect site. The bone threshold range includes some voxels of the βTCP due to the above-mentioned artefacts; this was dealt with by later Boolean subtraction of the final βTCP segmentation. Meanwhile, the βTCP lattice threshold was set to 5300 and above to completely avoid inclusion of any bone in the βTCP segmentation. This value underestimated the known thickness of the βTCP struts in the segmentation and so a geometric offset of +200 μm was applied to compensate for the reduced thickness. The βTCP segmentation was then Boolean subtracted from the new bone segmentation to remove most of the erroneously included voxels near the surface of the βTCP. The remaining artefacts included in the bone features were used as reference points to align the μCT segmentations, the implant digital 3D models, and segmentations of the native bone from the CBCT taken immediately after surgery. This allowed for identification of the original boundary between the native mandibular bone and the new bone formed (Supplementary Fig. 11a & 11b). The volume of new bone formation at the defect site was split into regions within the βTCP lattice and the LS-PEK scaffold by Boolean operations with non-latticed versions of the implants and the original native bone and quantified using the segmentation statistics features of 3D Slicer® (Fig. 6).

## Mechanical testing

Mechanical tests were conducted sequentially on a TMA-WDW-10E Universal Testing Machine from Test Machines Australia (Melbourne, Australia). The test machine utilises a 10 kN load cell calibrated and verified to Class 0.5 ISO 376 and operated in a uniaxial loading mode. Mechanical testing of the full mandible used two high-tensile straps designed to simulate the pull of the medial pterygoid and masseter muscles whilst loading the condyle and incisor regions and allowing rotation of the condyle/coronoid. The LS-PEK frame was inset to the mandibular defect with Ti screws. The test was aimed to destructively assess the force at failure of the implant. Mechanical testing of the hemi-mandibles utilized two-piece sheep-specific moulds cast in epoxy resin (Epoxycast clear, Bankstown, Australia) to hold the condyle and coronoid of the mandible in position during testing (Supplementary Fig. 12). The screws that were used for fixation of the LS-PEK frame were removed prior to testing. The test was aimed to non-destructively assess the structural stiffness of the host bone–scaffold implant system. By carefully controlling the applied load, the overall integrity of the sample was maintained to make the subsequent histological assessment possible. An upward displacement was applied by using the loading cell (10 kN) at a velocity of 1 mm/min until the first signs of catastrophic fracture were observed.

## Histology analysis

Following mechanical testing, the hemi-mandibles were immediately fixed in 10% neutral buffered formalin at 4 °C for a maximum duration of one week. The specimens were fully immersed to ensure complete tissue penetration of the fixative. Subsequently, the samples were transferred to 70% ethanol before histological sample processing. Sheep mandible samples were trimmed with a precision band saw (EXAKT Advanced Technologies GmbH, Norderstedt, Germany) to remove excess bone tissues around the artificial bone, making sure the samples would fit on the 100 mm × 50 mm slides after embedding. The samples were dehydrated with increasing grades of ethanol and then infiltrated and embedded in Technovit 9100 methyl methacrylate system (Kulzer GmbH, Wehrheim, Germany) without decalcification[36]. Three resin ground sections were obtained per defect, at different depths within the scaffold sized at 80 mm × 45 mm at approximately 50 μm thickness: Section 1: showing the tissue regeneration within the LS-PEK frame; Section 2: showing the tissue regeneration on the βTCP lattice; and Section 3: showing the bone formation beyond the LS-PEK frame. The sections were collected on 100 mm × 50 mm glass microscope slides using an EXAKT cutting and grinding system (EXAKT Advanced Technologies GmbH, Norderstedt, Germany). The ground sections were stained using Goldner's trichrome[36]. The sections were stained with Weigert's haematoxylin (Merck, Bayswater, VIC, Australia) for 25 min then washed and immersed in acid Fuchsin-Ponceau working solution Fuchsin (Merck, Bayswater, VIC, Australia) for 10 min. Following washes in 1% acetic acid, the sections were stained with tungstophosphoric acid - orange G Fuchsin solution (Merck, Bayswater, VIC, Australia) for 20 min and light green solution for 15 min. After air drying, the sections were cleared in xylene and mounted for imaging. The images were captured with a Zeiss Observer 7 microscope (Carl Zeiss Microscopy, Germany) at x5 magnification via the brightfield tile acquisition mode. Images were captured with an Axiocam 512 camera using an EC Plan-Neofluar 5× objective lens (NA 0.16). Image acquisition, tiles stitching and processing were carried out with ZEN 3.1 (blue edition) software.

## Immunohistochemistry

Sheep resin sections were deplasticized in 3 changes of 2-Methoxyethyl acetate (2-MEA, Merck, Australia) and rehydrated through a graded series of ethanol to Tris-HCL buffer. Antigen retrieval was performed by incubating the sections with proteinase K for 10 min. Endogenous peroxidase activity was quenched with 3% hydrogen peroxide for 10 min. After wash in Tris-HCL buffer, the sections were blocked with 2% bovine serum albumin (BSA) and incubated with primary antibodies overnight at 4 °C. The following primary antibodies were used: anti-CD68 (ab125212, 1.67 μg/mL, Abcam), anti-CD206 (ab64693, 1.6 μg/mL, Abcam), and anti- iNOS (ab15323, 2 μg/mL, Abcam). After primary antibody incubation, sections were washed in Tris-HCL buffer and incubated with the HRP conjugated secondary antibody (EnVision™+ Dual Link System-HRP, product number: K4061, Agilent, Australia) for 2 h. Colour development was achieved using DAB (3,3'-diaminobenzidine). Slides were dehydrated, cleared, and mounted using a permanent mounting medium. Negative controls included omission of the primary antibody. The stained slides were imaged using a Zeiss Axio Observer 7 microscope in brightfield mode. Images were captured with an Axiocam 512 camera using a Plan-Apochromat 20× objective lens (NA 0.8). Image acquisition and processing were carried out with ZEN 3.1 (blue edition) software.

## Immunofluorescence

Sheep resin sections were deplasticized and rehydrated to Tris-HCL buffer. Antigen retrieval was performed by incubating the sections with proteinase K for 10 min. The sections were washed in Tris-HCL buffer and permeated with 0.2% Triton-X (in PBS) for 5 minutes. Then the sections were blocked with 2% BSA for 1 h at room temperature to reduce non-specific binding. Sections were then incubated overnight at 4 °C with a ready-to-use rabbit polyclonal anti-von Willebrand

Factor (vWF) antibody (product number: GA527, Agilent). Following PBS washes, slides were incubated for 2 h at room temperature with an Alexa Fluor™ 488-conjugated goat anti-rabbit IgG (H + L), cross-adsorbed secondary antibody (Thermo Fisher Scientific, Australia). After final washes, nuclei were counterstained with DAPI (Thermo Fisher Scientific, Australia). Slides were mounted with antifade fluorescence mounting medium and imaged using an Olympus FV4000 confocal microscope (Evident, Australia). Negative controls were prepared by omitting the primary antibody. Imaging was performed using an Olympus FV4000 confocal microscope (Evident, Australia) equipped with the EVIDENT® 1394 digital camera system. Images were acquired using a UPLXAPO 10× objective lens (magnification 10.0×, numerical aperture 0.4) with a 10% laser ND filter. Z-stack imaging was performed across the full section thickness, and images were reconstructed by maximum intensity projection. Acquisition and post-processing were carried out using cellSens FV software (Evident).

## Statistical analysis

Statistical analysis was carried out using GraphPad Prism 10.4.1 (GraphPad Software Inc., San Diego, CA, USA). The data are presented as mean ± standard deviation (SD) or mean ± standard error of the mean (SEM). We compared groups using paired two-tailed t tests, one-tailed Mann–Whitney U tests, and repeated-measures one-way analysis of variance (ANOVA), and used linear and non-linear quadratic least squares regression analyses to examine relationships in bone growth, scaffold degradation, mechanical testing, and force–displacement curves.

## Reporting summary

Further information on research design is available in the Nature Portfolio Reporting Summary linked to this article.

## Data availability

The data that support the findings of this study are available within this article and its Supplementary Information and source data files. Source data are provided with this paper. All data supporting the finding is also available from the corresponding authors upon request. Source data are provided with this paper.

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

## Acknowledgements

Funding for this work was provided by The Cancer Institute New South Wales (CINSW 2020/2081 JRC), Sydney Local Health District (JRC), The Lang Walker Family Foundation (JRC), Surfebruary (DAM), The National Health and Medical Research Council (2024602 JRC, JMC, QL, MW, GW, RG), Arto Hardy Family (JMC), Garnett Passe Conjoint Grant 2023 (DAM), The Ian Potter Foundation (JMC), and Australian Government (JMC). The authors acknowledge the facilities and the scientific and technical assistance (Lisa Partel) of Hybrid Theater- Sydney Imaging and Pre Clinical Imaging, Core Research Facility, University of Sydney, and Linda Rogers for her assistance with immunohistochemistry analysis included in Fig. 2. We also acknowledge the Central Analytical Research Facility (CARF, QUT) for the resin histology preparation, analysis and microscopic imaging. The authors also thank Australian National Fabrication Facility (ANFF) and Sydney Manufacturing Hub (University of Sydney) for the manufacturing of βTCP lattice used in the artificial bone. BioRender.com was used to create Fig. 1.

## Author contributions

J.R.C., D.A.M., D.R.M., G.W., E.T.C and J.M.C. conceived the project. J.R.C., D.A.M., D.R.M., K.C., E.T.C., J.M.C., W.T.L., and D.K.L. designed the devices for the project. K.C., D.K.L., and W.T.L. fabricated devices for the project. K.C., W.T.L., B.W., H.X., D.A.M., E.T.C. and A.G. performed benchtop testing. J.R.C., D.A.M., A.G., D.L., T.M., K.P., J.W., C.F., D.H., and T.H.L. designed the surgical procedure and conducted surgeries, postoperative animal monitoring/care (with assistance from I.W.). D.A.M., H.X., A.G., K.C., M.A.W., and J.R. performed histology sample preparation, analysis of quantitative histology (with assistance from R.G.), conebeam- and micro-computed tomography collection, and video collection. E.T.C. isolated, expanded, and cultured stem cells (with assistance from M.A.R. and D.A.M.), and assembled the stem cell-infused ceramic lattice. D.R.M. and H.V.K. performed plasma treatment. B.W., C.W., and Q.L. performed the biomechanical computational analyses. D.A.M., J.R.C., J.R., B.W., E.T.C, J.M.C. and C.W. analysed the data. J.R.C., J.M.C., D.A.M., E.T.C., D.L., H.V.K., W.T.L., J.R., and A.G. wrote the paper. All authors read and provided comments on the paper.

## Competing interests

J.M.C., J.R.C., D.R.M., W.T.L., D.H., D.A.M., H.V.K., K.C., D.L. and E.T.C. are co-inventors on a patent application related to the technology discussed in this study (Australian Provisional Patent Application No. 2023903710, filed November 2023, and International Patent Application No. PCT/AU2024/051213, filed November 2024). The patent application is entitled "Implants for teeth and bone", was filed by Chris O'Brien Lifehouse, and pertains to the manufacture and use of the non-resorbable artificial bone technology described in the manuscript. The other authors declare no competing interests.

## Additional information

[1]NHMRC Centre of Research Excellence for Applied Innovations in Oral Cancer, Camperdown, NSW, Australia. [2]Integrated Prosthetics and Reconstruction, Chris O'Brien Lifehouse, Camperdown, NSW, Australia. [3]Central Clinical School, Faculty of Medicine and Health, The University of Sydney, Camperdown, NSW, Australia. [4]Royal Prince Alfred Institute of Academic Surgery, Sydney Local Health District, Camperdown, NSW, Australia. [5]Department of Head and Neck Surgery, Sydney Head and Neck Cancer Institute, Chris O'Brien Lifehouse, Camperdown, NSW, Australia. [6]Arto Hardy Family Biomedical Innovation Hub, Chris O'Brien Lifehouse, Camperdown, NSW, Australia. [7]Intelligent Polymer Research Institute and Institute of Innovative Materials, Faculty of Engineering and Information Sciences, The University of Wollongong, Fairy Meadow, NSW, Australia. [8]School of Medical Sciences, Faculty of Medicine and Health, The University of Sydney, Camperdown, NSW, Australia. [9]Sarcoma and Surgical Research Centre, Chris O'Brien Lifehouse, Camperdown, NSW, Australia. [10]School of Aerospace, Mechanical and Mechatronic Engineering, The University of Sydney, Camperdown, NSW, Australia. [11]ARC Training Centre for Cells and Tissue Engineering Technologies, Central Analytical Research Facility, Research Infrastructure, Queensland University of Technology, Brisbane, QLD, Australia. [12]School of Physics, The University of Sydney, Camperdown, NSW, Australia. [13]Australian National Fabrication Facility-Materials Node, AIIM Facility, The University of Wollongong, Wollongong, NSW, Australia. [14]Laboratory Animal Services, The University of Sydney, Camperdown, NSW, Australia. [15]School of Mechanical, Medical and Process Engineering, Queensland University of Technology, Brisbane, QLD, Australia. [16]Department of Otolaryngology –

Head & Neck Surgery, Faculty of Medicine and Health Sciences, Macquarie University, Macquarie Park, NSW, Australia. [17]Sydney Dental School, Faculty of Medicine and Health, The University of Sydney, Camperdown, NSW, Australia. [18]Department of Tissue Pathology and Diagnostic Oncology, Royal Prince Alfred Hospital, Camperdown, NSW, Australia. [19]Centre for Advanced Materials Technology, The University of Sydney, Darlington, NSW, Australia. ✉e-mail: jonathan.clark@lh.org.au; jeremy.crook@lh.org.au

