## [Transparent Peer review file · Nature Communications]

Mechanobiologically-optimized non-resorbable artificial bone for patient-matched scaffold-guided bone regeneration

Corresponding Author: Jonathan Clark

Version 0:

Reviewer comments:

Reviewer #1

(Remarks to the Author)

What are the noteworthy results?

Answer: In this paper, the authors developed a patient-specific 3D-printed polyetherketone gyroid scaffold, which was successfully implanted to reconstruct segmental mandibular defects in sheep. The scaffold integrates a resorbable ceramic lattice and a stem cell-laden hydrogel, serving as an osteoinductive reservoir. The results demonstrated excellent clinical performance and bone fusion, suggesting its potential as a viable alternative to conventional bone grafting and metallic plate fixation.

Will the work be of significance to the field and related fields? How does it compare to the established literature? If the work is not original, please provide relevant references

Answer: The reviewer recommends that the authors remove expressions suggesting that this study is the "first" of its kind, as there are existing cases of using PEK for bone replacement and in vivo implantation for mandibular defects. Such claims should be avoided in research papers.

Does the work support the conclusions and claims, or is additional evidence needed?

Answer: The reviewer recommends that the authors restructure the Results section to ensure that it presents only experimental findings, as it currently contains numerous discussions that should be placed in the Discussion section. The authors should separate and rewrite these sections accordingly to maintain clarity and proper organization of the manuscript.

Are there any flaws in the data analysis, interpretation and conclusions? Do these prohibit publication or require revision?

Answer: The reviewer recommends that the authors perform immunohistochemistry or fluorescence staining using bone-specific markers to confirm whether the proliferating cells after implantation have successfully differentiated into bone cells. Additionally, fibrosis-specific markers should be used to assess whether these bone cells maintain a healthy state. The reviewer notes that the paper contains an excessive amount of simulation results. To enhance the credibility of the functional assessment, the authors are advised to minimize reliance on simulation data and instead prioritize presenting experimental results, such as staining images of the implanted tissue sections, PCR data, or blood test results.

Is the methodology sound? Does the work meet the expected standards in your field?

Answer: The reviewer observes that the experimental data across sheep A–E show significant variability, making it difficult to draw statistically meaningful conclusions. Instead of comparing individual experimental animals, the authors are advised to include positive and negative control groups (normal sheep and sheep with defect creation but without scaffold implantation). Comparing results with these control groups would provide more meaningful insights and better demonstrate the superiority of the scaffold.

Is there enough detail provided in the methods for the work to be reproduced?

Answer: The reviewer observes that, based on the post-implantation results, the rate of cell proliferation appears to be slower than the degradation rate of the scaffold. As a result, the scaffold may have degraded before being sufficiently replaced by bone cells. The authors should clearly explain whether this outcome affects the functional integrity of the regenerated bone. Additionally, the reviewer recommends including actual images of the excised bone and modifying the figures to clearly distinguish the original bone from the scaffold location. This will help provide an intuitive understanding of the expected bone regeneration process.

Original reviewer's comment

In this paper, the authors developed a patient-specific 3D-printed polyetherketone gyroid scaffold, which was successfully implanted to reconstruct segmental mandibular defects in sheep. The scaffold integrates a resorbable ceramic lattice and a stem cell-laden hydrogel, serving as an osteoinductive reservoir. The results demonstrated excellent clinical performance and bone fusion, suggesting its potential as a viable alternative to conventional bone grafting and metallic plate fixation. For this manuscript to be improved, the reviewer would like to make comments as follows:

1. There were many typos and grammatical errors, so please correct them. In some cases, the abbreviated form of a term is inconsistent, or the full name is not provided. Therefore, the reviewer recommends that the authors revise all cases for consistency and clarity.
2. The reviewer recommends that the authors remove expressions suggesting that this study is the "first" of its kind, as there are existing cases of using PEK for bone replacement and in vivo implantation for mandibular defects. Such claims should be avoided in research papers.
3. The reviewer recommends that the authors expand the introduction to include a broader discussion of various biofabrication techniques, including 3D bioprinting for bone defect reconstruction, as well as multilayered scaffold fabrication incorporating vascular and musculoskeletal functional structures. Additionally, the authors should introduce and describe the advantages of other biomaterials before clearly explaining why the selected SLS printing technique and biomaterial used in this study are distinctive. This will help emphasize the novelty of the proposed approach.
4. The reviewer recommends that the authors verify whether the viability of cells incorporated into the implanted structure is affected when using the selected SLS printing technique. Given that photoinitiators and other biomaterials used in SLS printing are known to exhibit cytotoxicity, it is essential to confirm that the employed method does not compromise cell viability.
5. The reviewer recommends that the authors restructure the Results section to ensure that it presents only experimental findings, as it currently contains numerous discussions that should be placed in the Discussion section. The authors should separate and rewrite these sections accordingly to maintain clarity and proper organization of the manuscript.
6. The reviewer recommends that the authors include a schematic illustrating the entire fabrication process and provide a video demonstrating the scaffold fabrication, as the current figures alone are insufficient for a clear understanding of the process. Additionally, the authors should specify the maximum and minimum dimensions of the scaffold that can be fabricated. Demonstrating its applicability to various bone defect sites beyond the mandible would further highlight the superiority and versatility of the proposed scaffold.
7. The reviewer notes that inflammation is a critical issue in scaffold implantation, yet no direct experiments addressing this concern appear to have been conducted in the study. Therefore, the authors are recommended to select two to three immune-specific markers and perform qualitative analyses, such as immunohistochemistry or fluorescence staining, along with quantitative analyses, such as PCR, to clearly validate the expression of immune-specific markers in the implanted tissue sections.
8. The reviewer recommends that the authors include p-values in all graphs, as they are currently missing. Adding p-values to all data will enhance the reliability and statistical significance of the results.
9. The reviewer observes that, based on the post-implantation results, the rate of cell proliferation appears to be slower than the degradation rate of the scaffold. As a result, the scaffold may have degraded before being sufficiently replaced by bone cells. The authors should clearly explain whether this outcome affects the functional integrity of the regenerated bone. Additionally, the reviewer recommends including actual images of the excised bone and modifying the figures to clearly distinguish the original bone from the scaffold location. This will help provide an intuitive understanding of the expected bone regeneration process.
10. The reviewer recommends that the authors excise both the native tissue and the implanted region together to assess whether the integration of pre-existing vasculature and muscle was successfully achieved. Immunohistochemistry or fluorescence staining should be performed on the excised tissue sections to provide clear evidence of successful tissue integration.
11. The reviewer recommends that the authors perform immunohistochemistry or fluorescence staining using bone-specific markers to confirm whether the proliferating cells after implantation have successfully differentiated into bone cells. Additionally, fibrosis-specific markers should be used to assess whether these bone cells maintain a healthy state.
12. The reviewer notes that the paper contains an excessive amount of simulation results. To enhance the credibility of the functional assessment, the authors are advised to minimize reliance on simulation data and instead prioritize presenting experimental results, such as staining images of the implanted tissue sections, PCR data, or blood test results.
13. The reviewer observes that the experimental data across sheep A–E show significant variability, making it difficult to draw statistically meaningful conclusions. Instead of comparing individual experimental animals, the authors are advised to include positive and negative control groups (normal sheep and sheep with defect creation but without scaffold implantation). Comparing results with these control groups would provide more meaningful insights and better demonstrate the superiority of the scaffold.
14. The reviewer notes that various biofabrication techniques have been developed to simultaneously construct compartmentalized structures incorporating vascular, musculoskeletal, and extracellular matrix (ECM) environments. In contrast, the scaffold presented in this study requires multiple fabrication steps and utilizes only osteogenic cells, limiting its applicability to simultaneous regeneration of multiple tissues such as vasculature and muscle. Therefore, the authors must clearly articulate the unique advantages of their scaffold. Additionally, many other research teams have already developed vascularized bone scaffold using stem cells and GelMA. Therefore, achieving results by fabricating single-cell-type regenerative scaffold using existing biomaterials and cell sources does not provide sufficient novelty for the paper. Consequently, the findings of this paper are neither particularly groundbreaking nor significantly novel. For these reasons, the reviewer concludes that this paper does not align with the desired scope of Nature Communications.

Reviewer #2

(Remarks to the Author)

Dear Authors and Editor:

I have reviewed the manuscript titled "Mechanobiologically-optimized non-resorbable artificial bone – a new paradigm in patient matched scaffold-guided bone regeneration". I appreciate the opportunity to assess this work. The study presents a novel and valuable approach, with in silico, in vitro, and in vivo testing, for bone reconstruction and stress shielding reduction with the introduction of a polymeric construct promoting bone growth, without the use of metallic or autologous components. The literature review is good, and the methodology used is sound. The results contribute to a deeper understanding of mandibular bone reconstruction. However, several areas need addressing to improve its clarity, rigor, and overall quality.

I recommend major revisions for this manuscript. I have provided my detailed comments and recommendations below.

Abstract

Line 74 – "Patient-matched scaffold guided bone regeneration (SGBR)"

The title has a hyphen between "scaffold" and "guided". Please homogenize along the text.

Line 76 – "However, translation has been hampered by the focus on bioresorbable scaffolds where the rate of scaffold degradation needs to be precisely and predictably matched to the rate of new bone formation."

Needs to be rewritten. What does the author refer to with "translation"? The word "scaffold" is repeated; use another term or rewrite.

Line 79 – Please rewrite to state that off-the-shelf, gold standard, clinically used, or commercially available metallic plates promote stress shielding.

Introduction

Line 94 – "It is estimated that more than 1.6 million bone grafts are used to treat such defects in the U.S. each year, costing a staggering USD\$244 billion."

Does this number include all types of bone grafts used in critical-sized defects?

Line 96 – "Segmental defects of the mandible created in the treatment of cancer are one of the most challenging to manage due to the high aesthetic and functional importance of the jaw, the extreme repetitive tensile and shear stresses associated with mastication, and the frequent use of radiotherapy."

Segmental defects are also created during reconstruction surgery after trauma, not only after cancer. Please include this information unless it is important to draw attention to the cancer treatment.

Line 102 – Please also add other Ti alloys (Ti64) and other metallic materials used clinically (SS316L).

Line 147 – "...however, it has never been used for segmental bone defect repair."

Has PEK been used for other implantable medical devices? Please include references.

Line 148 – "PEK can be selectively laser-sintered (SLS)"

Please use the correct terminology for the AM technology according to ISO/ASTM 52900-15. Powder Bed Fusion using Laser Sintering (PBF-LS). Homogenize for the entire manuscript.

Line 171 – From line 171 through the end of the introduction section, it looks more like an abstract than an introduction section. Please reduce the information. Just state what the approach you followed was and state the main result in one sentence.

Please include references to other stiffness-matched approaches (more elastic materials and porous implants) for metallic implants and what problems this approach tackles. For example:

Khatab, N.R., Olivas-Alanis, L.H., Chmielewska-Wysocka, A. et al. Evaluation of stiffness-matched, 3D-printed, NiTi mandibular graft fixation in an ovine model. *BioMed Eng OnLine* 23, 105 (2024). <https://doi.org/10.1186/s12938-024-01289-x>

Results

Line 194 – "replicate selective laser melted (SLM) grade 23"

Please use the correct terminology for the AM technology according to ISO/ASTM 52900-15. Powder Bed Fusion using Laser Beam (PBF-LB). Homogenize for the entire manuscript.

Line 268 – Figure 3.A

Indicate what views of the implant are shown in the image (top, bottom, cross section...)

Also, show the structure used for the TCP structure.

Line 268 – Figure 3.B

Why does the implant shown in the mechanical setup have an incomplete porous structure?

Why is the model restrained at the front teeth? The boundary conditions seem different for this setup than for the implanted bones.

Line 268 – Figure 3.C

Why is there a gap between the porous gyroid region and the solid "shell"? Figure 3A does not show this gap, and it shows

that the solid and porous parts are joined.

Line 304 – “ μ CT demonstrated osteogenesis through the gyroid structure ranging from 1.16 cm³ to 2.65 cm³ in some cases, bridging the entire bone defect.”

Would it be useful to relate these numbers to the porous volume of the implant? What's the meaning of this 1.16-2.65 cm³ volume compared to the defect?

Line 362 – “The shift of peak VM stress to the mandibular bone reduces the risk of 363 ductile-related damage, such as yielding and fatigue failure.

Please rewrite for better understanding.

Line 388 – Figure 6

What is the meaning of the cracks on the host mandible? For me, they were promoted by how the mechanical testing setup was designed. It will be more useful to have a crack initiator site or grow information for the construct or the host bone (away from the resin fixation areas). Like the information presented in Figure 3.B

Line 389 – Caption Figure 6

Please use F(i), F(ii), G(i), G(ii) in the caption, or change the figure to f1,f2,g1,g2.

Methods

Line 503 – “The angle of the mandible was selected because it is a site of maximum stress”,
Under what loading? Unilateral? Frontal? Please include references.

Line 523 – “The 523 scaffold's pore structure is composed of 1 mm pores arranged in a 3D array of periodic unit cells 524 based on Schwarz P-surfaces.”

What's the strut size? What's the resulting porosity? How did you measure/define the strut and pore size?

Line 533 – “The SLS-PEK frame was custom designed to fit each sheep's left mandible using a clinically validated VSP process where the ablation and reconstruction are digitally simulated.”

Please add a reference for the clinically validated VSP process.

Line 548 – “The middle section bridges the bone defect employing a single gyroid TPMS scaffold architecture (3.5 mm unit size and 1.6 mm porosity) with solid reinforcement superiorly and inferiorly.”

What's the porosity level (%)? Why was it changed from 1 mm to 1.6 mm? Please replace “1.6mm porosity” with “1.6 pore size”. If you haven't done it yet, define how you measured the pore size. What about the strut size? How were these parameters selected? It is stated that it is based on numerical modelling, but were other sizes evaluated? Please include references if useful.

Line 554 – “The lattice cell size was selected to be 1.875 mm, with a beam thickness of 0.4 mm”

Define what lattice cell was used. And how did you measure the pore and strut size? (if different than TPMS Gyroid). Was it the one in line 563? If so, please rewrite to include that information in line 554.

Line 558 – “The design process used specialized lattice generation software to create 559 a lattice structure within the confines of the desired geometry to ensure that the lattice would 560 align with the boundaries of the planned insertion void.”
What software was used for the generation of the porous structure?

Line 575 – What are “2,ss elements”? Please correct, rewrite, or define.

Line 587 – “Two osseointegration conditions were considered: the first scenario assumed a lack of osseointegration, while the second scenario mimicked the presence of osseointegration between the scaffold and the host mandible.”

Is the second condition assuming bone ingrowth? Or it is only considering a tied bone between the host bone and the porous implant.

Line 614 – “The load and boundary conditions were set to simulate the biting forces for the sheep mandible”

What are those boundary conditions?

Why wasn't it considered a chewing model for the evaluation of the construct?

Line 637 - Was there any morphological assessment of the implants after printing (SLS and DLP) or postprocessing? How different are they from design?

Line 820 – “The volume of new bone formation at the defect site was quantified using the segmentation statistics feature of 3D Slicer”

How do you confirm that you are comparing the same volume/area? Are there any boundaries?

Line 826 – “Mechanical testing of the full mandible used two high-tensile straps designed to simulate the pull of the medial pterygoid and masseter muscles whilst loading the condyle and incisor regions and allowing rotation of the condyle/coronoid.”

How is the stiffness of the straps related to the muscles' properties?

Version 1:

Reviewer comments:

Reviewer #1

(Remarks to the Author)

In this paper, the authors developed a patient-specific 3D-printed polyetherketone gyroid scaffold, which was successfully implanted to reconstruct segmental mandibular defects in sheep. The scaffold integrates a resorbable ceramic lattice and a stem cell-laden hydrogel, serving as an osteoinductive reservoir. The results demonstrated excellent clinical performance and bone fusion, suggesting its potential as a viable alternative to conventional bone grafting and metallic plate fixation.

1. The reviewer judge that most of the comments previously provided by the reviewer were accepted by the authors and revised accordingly. However, the reviewer recommends that the authors make further revisions to address the remaining shortcomings in the manuscript. For this manuscript to be improved, the reviewer would like to make comments as follows:

2. All graphs should include p-values. As there are still figures in which p-values are missing, the reviewer recommends that the authors carefully review all graphs and ensure the inclusion of p-values to enhance the reliability and credibility of the results.

3. The manuscript still contains a substantial amount of simulation data, and it remains difficult to clearly distinguish between the native bone and the scaffold following implantation. Therefore, the reviewer recommends that the authors explicitly indicate, for each figure, which colors correspond to bone and scaffold, and use arrows within the images to clearly highlight the location of newly formed bone.

4. Because sheep exhibit highly active autologous bone regeneration, the most critical factor in validating the efficacy of the scaffold presented in this study is to clearly demonstrate whether the implanted cells have successfully differentiated into bone cells. To this end, the authors should employ specific markers that distinguish implanted cells from host cells to directly verify that the implanted cells have proliferated within the scaffold and effectively undergone osteogenic differentiation. Particularly in this study, where the use of a control group with segmental mandibular defects without scaffold implantation is not feasible due to ethical constraints, such analysis is considered essential to assess the contribution of the implanted cells. Considering the standard expected for a high-impact journal like Nature Communications, stating that these experiments will be addressed in future work appears insufficient to substantiate the scaffold's functional efficacy. Therefore, the reviewer strongly recommends that the authors perform the necessary experiments and present both quantitative and qualitative data to support their claims. In addition, the reviewer recommends that the authors explicitly write that, for ethical reasons, a control group with unreconstructed segmental mandibular defects could not be included in the study.

5. With regard to the fabrication of scaffolds incorporating functional vascular and musculoskeletal structures for bone defect repair, the reviewer finds that the introduction section still lacks sufficient discussion of existing biofabrication techniques and alternative biomaterials. Therefore, the reviewer recommends that the authors expand the introduction to include a more comprehensive overview of relevant biofabrication methods and biomaterials. Additionally, the authors should clearly explain why the specific SLS printing technique and biomaterial used in this study are superior compared to these alternatives, thereby emphasizing the novelty and significance of their approach.

6. In relation to the comments, the reviewer recommends that the authors refer to the two papers below and add the following papers as related references.

(1) Kim, Joeng Ju, and Dong-Woo Cho. "Advanced strategies in 3D bioprinting for vascular tissue engineering and disease modelling using smart bioinks." *Virtual and Physical Prototyping* 19.1 (2024): e2395470.

(2) Rifai, Aaqil, et al. "Biofabrication of functional bone tissue: defining tissue-engineered scaffolds from nature." *Frontiers in Bioengineering and Biotechnology* 11 (2023): 1185841.

Reviewer #2

(Remarks to the Author)

The reviewer appreciate the author's time and effort to attend the comments and improve the quality of the work. I consider this manuscript is ready for publication with no further edits.

Version 2:

Reviewer comments:

Reviewer #1

(Remarks to the Author)

In this paper, the authors developed a patient-specific 3D-printed polyetherketone gyroid scaffold, which was successfully implanted to reconstruct segmental mandibular defects in sheep. The scaffold integrates a resorbable ceramic lattice and a stem cell-laden hydrogel, serving as an osteoinductive reservoir. The results demonstrated excellent clinical performance

and bone fusion, suggesting its potential as a viable alternative to conventional bone grafting and metallic plate fixation.

The reviewer would like to give authors an 'accept', because the reviewer think that authors have reflected and modified the previous our comment.

Dear Reviewers,

We thank you for taking the time to consider our manuscript and for the valuable suggestions. Below we provide a point-by-point response to the comments listed in italics. Please note that where reviewer 1's comments are duplicated, we have answered once for brevity, referring to our answer for subsequent responses.

Reviewer #1:

1. In this paper, the authors developed a patient-specific 3D-printed polyetherketone gyroid scaffold, which was successfully implanted to reconstruct segmental mandibular defects in sheep. The scaffold integrates a resorbable ceramic lattice and a stem cell-laden hydrogel, serving as an osteoinductive reservoir. The results demonstrated excellent clinical performance and bone fusion, suggesting its potential as a viable alternative to conventional bone grafting and metallic plate fixation.

Response: The authors thank the reviewer for their thoughtful comments.

2. The reviewer recommends that the authors remove expressions suggesting that this study is the "first" of its kind, as there are existing cases of using PEK for bone replacement and in vivo implantation for mandibular defects. Such claims should be avoided in research papers.

Response: Whilst we have now removed these claims from the manuscript, we believe they were accurate and were critical in order to highlight the novel nature of the research. Importantly, we have repeated our review of the literature and are not aware of any *in vivo* studies of laser sintered (LS)-PEK for bone replacement other than our own. If the reviewer is aware of any literature relevant to their claim, we would like to be made aware of the publication(s) in order to cite them in the manuscript. Further, there are no published studies of reconstruction of segmental mandibular defects in large animal models without metal plate fixation. Sillmann et al (reference below) provide an excellent review of animal models used for mandibular reconstruction. The reviewer will note that all segmental defects listed in Table 1 of Sillmann et al have relied on metal plate fixation for success and that the defect created in our study is almost twice the length of all other segmental defects, making it the only study that is clinically relevant to oncological surgery.

The clinical importance of our innovation is explained in the following text:

(P4, Lines 127 – 141) “This morbidity includes the sacrifice of muscles, nerves, and blood vessels during bone harvest and is exacerbated by subsequent complications of metal plate fixation, requiring plate removal in approximately 25% of patients. The high elastic modulus of Ti and other metals causes both mechanical stress shielding and unwanted stress concentrations, which may contribute to bone resorption and osteolysis, impairing bone healing and increasing the likelihood of fixation hardware failure. The stress shielding associated with high-modulus materials may be lessened by ‘stiffness matching’, which can be achieved by changing material composition through the use of composites or by changing its geometry, i.e., location, shape or porosity. Unfortunately, this is not always feasible due to anatomical constraints, and whilst some problems may be overcome by optimizing the mechanical properties, this does not address issues related to X-ray perturbation during postoperative radiotherapy and associated imaging artifacts. X-ray perturbation can increase the dose of radiotherapy to healthy bone, underdose the target volume, and interfere with surveillance imaging modalities, impeding early detection of infection and local recurrence of cancer (Fig. 1).”

Note, we are concerned that the reviewer may have confused polyetherketone (PEK) with other polyaryletherketone (PAEK) family members, polyetherketoneketone (PEKK) or polyetheretherketone (PEEK). This is complicated by the commercial labelling of PEK as EOS HP3 PEEK. The EOSINT P800 printer is also able to print PEKK (Oxford Performance Materials PEKK CT, USA). We also comment on this issue in response to Reviewer 2 (please see response to point 8).

Reference: Sillmann YM, Eber P, Orbeta E, Wilde F, Gross AJ, Guastaldi FPS. Milestones in Mandibular Bone Tissue Engineering: A Systematic Review of Large Animal Models and Critical-Sized Defects. *J Clin Med.* 2025 Apr 15;14(8):2717. doi: 10.3390/jcm14082717. PMID: 40283548; PMCID: PMC12027812.

3. *The reviewer recommends that the authors restructure the Results section to ensure that it presents only experimental findings, as it currently contains numerous discussions that should be placed in the Discussion section. The authors should separate and rewrite these sections accordingly to maintain clarity and proper organization of the manuscript.*

Response: We have restructured the results section to remove elements of discussion.

4. The reviewer recommends that the authors perform immunohistochemistry or fluorescence staining using bone-specific markers to confirm whether the proliferating cells after implantation have successfully differentiated into bone cells. Additionally, fibrosis-specific markers should be used to assess whether these bone cells maintain a healthy state. The reviewer notes that the paper contains an excessive amount of simulation results. To enhance the credibility of the functional assessment, the authors are advised to minimize reliance on simulation data and instead prioritize presenting experimental results, such as staining images of the implanted tissue sections, PCR data, or blood test results.

Response: Regarding ascertaining whether the implanted cells have differentiated into bone cells, it would require a fluorescent tag to be incorporated prior to cell implantation to distinguish the implanted cells from the host cells. This step was unfortunately not undertaken and as such we cannot presently distinguish between implanted and host cells within the degradable ADSC-laden β TCP lattice (**P18, Lines 510 - 524**). Whilst we will consider doing so for future planned studies, for now, we reiterate from the manuscript that the primary purpose of infusing the osteogenically differentiated ADSCs was to enhance the bioavailability of calcium from the β TCP scaffold to facilitate native bone ingrowth and osseointegration of the LS-PEK implant. Based on the μ CT and histology data, we have strong evidence that successful osseointegration of the LS-PEK implants was achieved due to new bone growing into the voids of the LS-PEK gyroid structure.

As requested, (here and in comments 15 and 19), we have performed immunohistochemistry and immunofluorescence to look for evidence of vascularization and inflammation (See comment 15). Markers of inflammation, including key macrophage markers: CD68 (pan-macrophage marker, ab125212), CD206 (M2-like, anti-inflammatory marker, ab64693), and iNOS (M1-like, pro-inflammatory marker, ab15323) have shown negligible levels of unwanted inflammatory response. We can also report evidence for endothelial cells and blood vessels in both the β TCP lattice and LS-PEK implant. To reiterate from the manuscript, intense von Willebrand Factor (vWF) expression was observed at the interface between the host bone and the newly formed bone associated with each type of implant supporting new bone ingrowth (**P11, Lines 338 – 352**).

Regarding the simulation studies, the results are essential evidence supporting our principal tenant that the formation of bone in this study is predominantly driven by mechanical stress and strain as opposed to induced by the chemistry of the biomaterials.

They represent evidence of this mechanical concept in the same way that immunohistochemistry is evidence of cellular function. This is a fundamental departure from the use of bioresorbable materials and multilayer cellular constructs that are commonly being investigated by others but have failed to move beyond *in vitro* and small animal models to the clinically relevant segmental bone defects in large animal models that are presented in our manuscript.

5. The reviewer observes that the experimental data across sheep A–E show significant variability, making it difficult to draw statistically meaningful conclusions. Instead of comparing individual experimental animals, the authors are advised to include positive and negative control groups (normal sheep and sheep with defect creation but without scaffold implantation). Comparing results with these control groups would provide more meaningful insights and better demonstrate the superiority of the scaffold.

Response: As suggested by the reviewer we have added normal sheep as a control for the parameters of weight gain and masticatory rate (**P11, Lines 318 - 320; Figure 5Q; P17, Lines 468 - 471**). Normal sheep were maintained in the same facility and were fed with the same diet and gained weight at a similar rate to the implanted sheep.

Unfortunately, it is not possible to create a segmental defect in the mandible of sheep without reconstruction for humane reasons. The inability to perform statistical comparisons between the LS-PEK gyroid scaffolds and a control group is a significant but necessary limitation of the current methodology. The reviewer will note that there are no published studies where sheep have undergone segmental mandibulectomy without reconstruction. This issue was discussed extensively with the University of Sydney Veterinary team and the Animal Ethics Committee, who rejected the use of a control group without reconstruction on the grounds that the sheep would not be able to maintain nutrition. Ruminants cannot be tube-fed as they regurgitate food from their stomach to re-chew food (cud) before digestion. Thus, an unreconstructed defect would lead to rapid malnutrition and weight loss requiring termination of the study on humane grounds. Some studies have attempted to bridge bony defects with reconstruction plates alone. This has led to rapid plate and screw failure due to the high mechanical forces created during mastication, which sheep do for up to 8 hours per day. For these reasons the study focussed on demonstrating the feasibility of using our artificial bone to reconstruct segmental defects in the mandible

without metal plate fixation as opposed to comparing rates of bone formation with a negative control.

6. The reviewer observes that, based on the post-implantation results, the rate of cell proliferation appears to be slower than the degradation rate of the scaffold. As a result, the scaffold may have degraded before being sufficiently replaced by bone cells. The authors should clearly explain whether this outcome affects the functional integrity of the regenerated bone.

Response: We would like to clarify that there are two scaffold materials: 1. LS-PEK, which is permanent (i.e., it does not degrade), and 2. β TCP, which is bioresorbable (i.e., it does degrade) and was included as a calcium reservoir for osteoinduction and osteoconduction within the LS-PEK scaffold. We agree with the reviewer that the β TCP scaffolds degraded at highly variable rates (a key finding of this study). However, whilst in some cases degradation may have been too rapid for osteogenesis to occur directly on/within the β TCP scaffold, the LS-PEK scaffold demonstrated much larger amounts of new host bone ingrowth before there were visible signs of degradation of the β TCP scaffolds (Figure 5). Furthermore, although no sheep demonstrated sufficient new bone within the β TCP scaffold to support a biomechanically stable reconstruction of the defect (another key finding of this study), we posit that the functional integrity of the host bone ingrowth to the LS-PEK scaffold may have been augmented by the release of calcium (and phosphate) ions during β TCP scaffold degradation. Finally, the high variability of the β TCP degradation illustrates the challenges to clinical translation of using bioresorbable/degradable scaffolds where the timing of degradation must match the rate of new bone formation to provide structural integrity to the reconstruction. This problem is abrogated by our use of the permanent osteoconductive LS-PEK implant.

We have revised the text to include:

(P19, Lines 528-541) “Thus, the osteogenically differentiated ADSCs were infused to enhance the bioavailability of the calcium reservoir and host bone ingrowth may have been augmented by the release of calcium (and phosphate) ions during β TCP scaffold degradation. μ CT-based bone volume estimates may be unreliable because the β TCP lattice is radio-opaque and degradation products may be misinterpreted as new bone using μ CT. It is challenging to separate the thresholding limits for calcium which is present in both β TCP and in newly formed bone. To overcome this, we combined μ CT with histology to

provide more reliable quantification. The presence of new bone and neovascularization within the β TCP lattice could be identified histologically but we were unable to determine whether osteoblasts originated from the host or the ADSCs. Regardless, the volume of new bone generated within the β TCP lattice was insufficient for mechanical stability and the rate of β TCP degradation was highly variable. This confirmed our concern that bioresorbable scaffolds will always require permanent structural support unless the scaffold materials degrade at a rate that matches new bone formation during the healing process.”

7. The reviewer recommends including actual images of the excised bone and modifying the figures to clearly distinguish the original bone from the scaffold location. This will help provide an intuitive understanding of the expected bone regeneration process.

Response: The authors have added supplemental figures (**Supplementary Fig. S1**) showing detailed images of the surgery to clearly show what bone was excised. We have also modified **Fig. 6B** and **Supplemental Fig. S2** to more clearly define the defect so that the regeneration process is less ambiguous.

8. There were many typos and grammatical errors, so please correct them. In some cases, the abbreviated form of a term is inconsistent, or the full name is not provided. Therefore, the reviewer recommends that the authors revise all cases for consistency and clarity.

Response: The authors have corrected all identifiable grammatical errors and made abbreviations consistent throughout the manuscript.

9. The reviewer recommends that the authors remove expressions suggesting that this study is the "first" of its kind, as there are existing cases of using PEK for bone replacement and in vivo implantation for mandibular defects. Such claims should be avoided in research papers.

Response: Please refer to response point 2 above.

10. The reviewer recommends that the authors expand the introduction to include a broader discussion of various biofabrication techniques, including 3D bioprinting for bone defect reconstruction, as well as multilayered scaffold fabrication incorporating vascular and

musculoskeletal functional structures. Additionally, the authors should introduce and describe the advantages of other biomaterials before clearly explaining why the selected SLS printing technique and biomaterial used in this study are distinctive. This will help emphasize the novelty of the proposed approach.

Response: We have expanded the introduction to include a broader discussion of various biofabrication techniques and describe the potential advantages of other classes of biomaterials, though it is not feasible to provide a detailed discussion of each specific material (see below). Whilst we were unable to identify any *in vivo* studies of note using *multilayered scaffold fabrication incorporating vascular and musculoskeletal functional structures* for bone defect repair, we would be happy to include any references that the reviewer is able to provide.

The following has been added prior to explaining the distinctive advantages of LS-PEK:

(P4, Lines 142 - 159) “There is considerable interest in using SGBR in the form of patient-matched medical devices (PMMDs) that are additively manufactured to precisely match the defect size and shape and abrogate the need for autologous bone grafts. Despite this, SGBR has failed to deliver constructs that are suitable for use in clinical practice because most tissue engineering strategies employ bioresorbable scaffolds composed of polymers or ceramics that are progressively replaced by bone as they degrade. This concept is attractive; however, it relies (unrealistically) on synchronizing the rates of degradation and osteogenesis to achieve the desired mechanical properties in the medium- and long-term. Most bioresorbable scaffolds are designed to withstand compressive loads but require reinforcement with metal plates or intramedullary nails to address issues with fixation, bending moment, and torsional stability. In fact, there are neither preclinical large-animal studies nor clinical studies published where SGBR has been used to reconstruct segmental bone defects without metal plate augmentation. Even with rigid fixation, bioresorbable scaffolds will ultimately fail if the rate of degradation exceeds the rate of new bone formation. It is conceivable that by engineering complex biomimetic structures containing multiple tissue lineages (bone, muscle, and vasculature), a more functional construct could be deployed. However, additional complexity encumbers clinical translation due to regulatory constraints, greater unpredictability, and the unsolved problem of how the viability of large living structures with complex geometries can be maintained when implanted.”

11. *The reviewer recommends that the authors verify whether the viability of cells incorporated into the implanted structure is affected when using the selected SLS printing technique. Given that photoinitiators and other biomaterials used in SLS printing are known to exhibit cytotoxicity, it is essential to confirm that the employed method does not compromise cell viability.*

Response: Whilst the additive manufacturing techniques involving laser sintering and ceramic printing were not and cannot be applied directly to cells, we agree that if cytotoxic residues were present, they may affect cell viability. To ensure this is not the case, we have demonstrated through *in vitro* and *in vivo* experiments (shown in Figure 2) that neither the LS-PEK scaffold nor the β TCP lattice are cytotoxic.

To clarify, the SLS printing technique was used for fabrication of the LS-PEK scaffold and the LS-PEK scaffold did not contact any cellular material prior to implantation. In contrast, the β TCP lattice was 3D-printed and sintered in a furnace before being infused with adipose derived stem cells (ADSC)-laden GelMA (see Figure 3). The viability of single cell dissociated adipose derived stem cells (ADSC) was confirmed using trypan blue exclusion assay prior to mixing with GelMA and lithium phenyl-2,4,6-trimethylbenzoylphosphinate (LAP). The ADSC-laden GelMA solution was then infused into the β TCP lattice and photocrosslinking was performed at 405nm in the visible spectrum wavelength to minimise cytotoxic effects due to DNA damage and the water soluble photoinitiator LAP used at a low concentration (0.25%) to maintain cytocompatibility (Ref below – Nyugen et al). Characteristic cell morphology demonstrating viable ADSC within GelMA in the β TCP lattice was confirmed using light microscopy during subsequent *in vitro* culture in osteogenic media prior to surgical implantation.

Reference: Nguyen AK, Goering PL, Elespuru RK, Sarkar Das S, Narayan RJ. The Photoinitiator Lithium Phenyl (2,4,6-Trimethylbenzoyl) Phosphinate with Exposure to 405 nm Light Is Cytotoxic to Mammalian Cells but Not Mutagenic in Bacterial Reverse Mutation Assays. *Polymers (Basel)*. 2020 Jul 3;12(7):1489.

12. *The reviewer recommends that the authors restructure the Results section to ensure that it presents only experimental findings, as it currently contains numerous discussions that should be placed in the Discussion section. The authors should separate and rewrite these sections accordingly to maintain clarity and proper organization of the manuscript.*

Response: Please refer to response point 3 above. We have restructured the results section to remove elements of discussion.

13. *The reviewer recommends that the authors include a schematic illustrating the entire fabrication process and provide a video demonstrating the scaffold fabrication, as the current figures alone are insufficient for a clear understanding of the process.*

Response: We have created a schematic (see **Figure 3**) and video (see **Supplementary Video 1**), which complements Figure 1 by further demonstrating the entire fabrication process.

14. *Additionally, the authors should specify the maximum and minimum dimensions of the scaffold that can be fabricated. Demonstrating its applicability to various bone defect sites beyond the mandible would further highlight the superiority and versatility of the proposed scaffold.*

Response: The authors have added the maximum and minimum dimensions of the scaffold that can be fabricated.

(P23, Lines 728 - 734) "The EOS P800 has a maximum build volume of 700 x 380 x 560 mm. Accounting for 5% shrinkage, this allows for a maximum scaffold size of 666 x 362 x 533 mm, more than enough for most bone-based scaffold applications. Meanwhile, the minimum scaffold size is dependent on the design. The minimum feature size the P800 can produce is approximately 0.8 mm which is appropriate for all bones except those of the middle ear. The powder was stored in 45% humidity at 22°C until printing. The building parameters of the LS-PEK frames are listed in Supplementary Table 1."

15. *The reviewer notes that inflammation is a critical issue in scaffold implantation, yet no direct experiments addressing this concern appear to have been conducted in the study. Therefore, the authors are recommended to select two to three immune-specific markers and perform qualitative analyses, such as immunohistochemistry or fluorescence staining, along with quantitative analyses, such as PCR, to clearly validate the expression of immune-specific markers in the implanted tissue sections.*

Response: We have now performed immunohistochemistry for markers of inflammation including key macrophage markers: CD68 (pan-macrophage marker, ab125212), CD206 (M2-like, anti-inflammatory marker, ab64693), and iNOS (M1-like, pro-inflammatory marker, ab15323). The macrophage response characterized by these markers has been added to the manuscript as follows:

(P11; lines 341 - 351) “The level of inflammation was assessed by immunohistochemistry using key macrophage markers: CD68 (pan-macrophage marker, ab125212), CD206 (M2-like, anti-inflammatory marker, ab64693), and iNOS (M1-like, pro-inflammatory marker, ab15323). As shown in Supplementary Fig. S6, the macrophage response characterized by these markers demonstrated low overall inflammatory response to the implanted scaffolds. CD68 showed negligible positivity around β TCP implants and weak positivity near LS-PEK implants, indicating a low overall macrophage presence, but slightly more pronounced in LS-PEK sections. CD206 staining showed medium positivity around β TCP and low positivity around LS-PEK, suggesting a more prominent M2-like (anti-inflammatory or tissue-remodelling) response to β TCP. Finally, iNOS exhibited very weak positivity around both implant materials, indicating negligible M1-like (pro-inflammatory) macrophage activation.”

Taken together, immunohistochemistry demonstrates low overall inflammatory response to the implanted scaffolds. We were unable to perform PCR as this requires the tissues to be treated immediately on explantation with specialised solutions (trizol) in order to preserve the RNA. Rather, tissues were fixed immediately on explantation to prioritize important histological information; this fixation is unfortunately not compatible with subsequent PCR analysis.

16. *The reviewer recommends that the authors include p-values in all graphs, as they are currently missing. Adding p-values to all data will enhance the reliability and statistical significance of the results.*

Response: We have added p values to the graphs where comparisons have been made. This includes:

(1) Figure 2 for comparisons between LS-PEK and titanium bone-implant interfaces, reverse 3-point bend tests, and pull-out tests.

(2) Figure 5 for comparisons in change in new bone volume within the LS-PEK frame, degradation of the β TCP lattice, and sheep weight over time.

17. The reviewer observes that, based on the post-implantation results, the rate of cell proliferation appears to be slower than the degradation rate of the scaffold. As a result, the scaffold may have degraded before being sufficiently replaced by bone cells. The authors should clearly explain whether this outcome affects the functional integrity of the regenerated bone.

Response: Please refer to response point 6 above.

18. Additionally, the reviewer recommends including actual images of the excised bone and modifying the figures to clearly distinguish the original bone from the scaffold location. This will help provide an intuitive understanding of the expected bone regeneration process.

Response: Please refer to response point 7 above.

19. The reviewer recommends that the authors excise both the native tissue and the implanted region together to assess whether the integration of pre-existing vasculature and muscle was successfully achieved. Immunohistochemistry or fluorescence staining should be performed on the excised tissue sections to provide clear evidence of successful tissue integration.

Response: We can confirm that both native tissue and the implanted region were excised together. Specifically, following euthanasia the entire mandible was excised from the sheep and then the muscular attachments were carefully removed from the implant.

Supplemental Figure S4 has been added to show this process and demonstrate that the muscle was well integrated with the LS-PEK scaffold and β TCP lattice. The surgeons performing the excision found that there was no sign of any unwanted macroscopic inflammatory response, such as granulation tissue or suppuration and that the overlying musculature was densely adherent to the implant suggesting high levels of soft tissue integration in addition to osseointegration.

The following text has been added:

(P11, Lines 320 – 326) “At sacrifice there was no capsule formation, granulation tissue, seroma, or suppuration and the associated masticatory musculature was strongly adherent to the artificial bone during explantation (Supplementary Fig. S4). This soft-tissue integration was confirmed prior to histological analysis when the adjacent musculature had to be excised from the scaffold prior to mechanical testing and resin embedding. The clinical and macroscopic evidence of implant integration was then confirmed microscopically by the absence of any unwanted inflammatory response on histological assessment as detailed below (Fig. 5I).”

We have now performed immunofluorescence labelling of new blood vessels at the interface of the native tissue and scaffold periphery to identify vasculature. Immunofluorescence of von Willebrand Factor (vWF) revealed strong labelling in both LS-PEK and β TCP implant sections. The integration of the scaffold into the native tissue can clearly be distinguished by vessels transversing the implant-tissue interface.

(P11, lines 338 - 342) “Immunofluorescence staining for von Willebrand Factor (vWF), a marker of endothelial cells and blood vessels, revealed strong positive signals in both β TCP (Fig. 6Avi, Supplementary Fig. S5) and LS-PEK (Fig. 6Ax, Supplementary Fig. S5) implant sections. Specifically, intense vWF expression was observed at the interface between the host bone and the newly formed bone surrounding each type of implant.”

20. *The reviewer recommends that the authors perform immunohistochemistry or fluorescence staining using bone-specific markers to confirm whether the proliferating cells after implantation have successfully differentiated into bone cells. Additionally, fibrosis-specific markers should be used to assess whether these bone cells maintain a healthy state. The reviewer notes that the paper contains an excessive amount of simulation results. To enhance the credibility of the functional assessment, the authors are advised to minimize reliance on simulation data and instead prioritize presenting experimental results, such as staining images of the implanted tissue sections, PCR data, or blood test results.*

Response: Please refer to response point 4.

21. *The reviewer observes that the experimental data across sheep A–E show significant variability, making it difficult to draw statistically meaningful conclusions. Instead of comparing individual experimental animals, the authors are advised to include positive and*

negative control groups (normal sheep and sheep with defect creation but without scaffold implantation). Comparing results with these control groups would provide more meaningful insights and better demonstrate the superiority of the scaffold.

Response: Please refer to response point 5.

22. *The reviewer notes that various biofabrication techniques have been developed to simultaneously construct compartmentalized structures incorporating vascular, musculoskeletal, and extracellular matrix (ECM) environments. In contrast, the scaffold presented in this study requires multiple fabrication steps and utilizes only osteogenic cells, limiting its applicability to simultaneous regeneration of multiple tissues such as vasculature and muscle. Therefore, the authors must clearly articulate the unique advantages of their scaffold. Additionally, many other research teams have already developed vascularized bone scaffold using stem cells and GelMA. Therefore, achieving results by fabricating single-cell-type regenerative scaffold using existing biomaterials and cell sources does not provide sufficient novelty for the paper. Consequently, the findings of this paper are neither particularly groundbreaking nor significantly novel. For these reasons, the reviewer concludes that this paper does not align with the desired scope of Nature Communications.*

Response: We acknowledge that there are other approaches to achieving musculoskeletal repair. We have taken a unique approach that we believe is more amenable to clinical translation and a paradigm shift away from bioresorbable scaffolds and bone grafting that will always require the support of metallic plates.

Whilst we agree that neovascularisation is a critical element, the inclusion of multiple exogenously delivered cell types and/or compartmentalized structures targeting the components is likely to detract from the translatability of our research. Rather, our focus on mechanical stimulation of bone regeneration is a clear point of difference, that makes our approach advantageous compared to other methods and, crucially, *clinically viable*.

Whilst the reviewer is concerned about the number of cell types, we do not believe this should determine the novelty of the technique since our goal was not to develop a biomimetic scaffold. Reiterating from the manuscript (Introduction and Discussion: paragraphs 5 and 3, respectively) osteogenically differentiated ADSCs were used to make the calcium phosphate within the β TCP insert more bioavailable, serving as a long-term calcium reservoir to enhance osteogenic activity of native/endogenous bone tissue rather than as a source of cells for bone growth per se. This is supported by our data shown in

Figs 5 and 6 where strong autologous bone growth occurs mostly in the non-resorbable plasma treated LS-PEK gyroid scaffold for the duration of the study. Our data also suggest that the stress-driven osteogenesis and surface characteristics that promote osseointegration are most important, and that the inclusion of cellular material may be unnecessary and certainly a hindrance to clinical translation. We further posit that the plasma immersion ion implantation treatment and laser sintering of the PEK scaffold create conditions on the surface and within the gyroid scaffold that encourage vascularised and mineralised host bone ingrowth (i.e., the environment needed for fully vascularised and mineralised bone). Based on these findings, a first-in-human trial of plasma-treated LS-PEK has been initiated.

In summary, the key novel findings are:

1. Stress driven osteogenesis as demonstrated by our simulation data - a move away from cellular-induced osteogenesis showing that complex multi-cellular engineering approaches are not necessary; and
2. A mechanically optimised permanent scaffold that osseointegrates, is strong enough to withstand masticatory forces, and does not require metal plate fixation – a move away from bioresorbable scaffolds.

We iterate that this has never been achieved in any study of segmental mandibular reconstruction using a clinically-relevant animal model, highlighting that indeed our findings are of great importance in the field of mandibular reconstruction.

Reviewer #2:

1. I have reviewed the manuscript titled "Mechanobiologically-optimized non-resorbable artificial bone – a new paradigm in patient matched scaffold-guided bone regeneration". I appreciate the opportunity to assess this work. The study presents a novel and valuable approach, with in silico, in vitro, and in vivo testing, for bone reconstruction and stress shielding reduction with the introduction of a polymeric construct promoting bone growth, without the use of metallic or autologous components. The literature review is good, and the methodology used is sound. The results contribute to a deeper understanding of mandibular bone reconstruction. However, several areas need addressing to improve its

clarity, rigor, and overall quality. I recommend major revisions for this manuscript. I have provided my detailed comments and recommendations below.

Response: We thank the reviewer for their thoughtful comments and suggested improvements.

2. Abstract Line 74 – “Patient-matched scaffold guided bone regeneration (SGBR)”
The title has a hyphen between “scaffold” and “guided”. Please homogenize along the text.

Response: This has been corrected.

3. Line 76 – “However, translation has been hampered by the focus on bioresorbable scaffolds where the rate of scaffold degradation needs to be precisely and predictably matched to the rate of new bone formation.”

Needs to be rewritten. What does the author refer to with “translation”? The word “scaffold” is repeated; use another term or rewrite.

Response: **1.** ‘Translation’ in this context refers to the translation of fundamental basic science research into clinical practice. This has been changed to ‘translation into clinical practice’ to make the intention clear. **2.** The second use of the word ‘scaffold’ has been removed.

4. Line 79 – Please rewrite to state that off-the-shelf, gold standard, clinically used, or commercially available metallic plates promote stress shielding.

Response: This has been changed to:

(P3, Lines 96 -98) “Consequently, there are no published studies using SGBR for segmental defects without metal plates, which is problematic because commercially available metallic plates cause stress shielding and X-ray perturbation.”

5. Introduction Line 94 – “It is estimated that more than 1.6 million bone grafts are used to treat such defects in the U.S. each year, costing a staggering USD\$244 billion.”
Does this number include all types of bone grafts used in critical-sized defects?

Response: This refers to the estimated cost of bone grafts used in clinical practice for critical-sized bone defects. The data comes from O’Keefe & Mao (reference below). The publication does not state the specific types of bone grafts (i.e., autografts, allografts, xenografts, or bone graft substitute materials), however, the breakdown of costs is as follows:

- 15 million fractures annually (\$45B);
- 1.6 million patients with trauma with hospital admission (\$27B);
- 2 million osteoporotic fractures (\$24B);
- 500,000 knee, 350,000 hip replacements, and 90,000 revision arthroplasty procedures (\$30B);
- 300,000 spinal fusions, 20,000 revision spine fusions (\$18B);
- 2400 primary bone malignancies; and 4500 benign tumors (\$100M).

Reference: O’Keefe & Mao. Bone tissue engineering and regeneration: From discovery to the clinic-an overview. Tissue Engineering - Part B: Reviews

<https://doi.org/10.1089/ten.teb.2011.0475>.

6. Line 96 – “Segmental defects of the mandible created in the treatment of cancer are one of the most challenging to manage due to the high aesthetic and functional 98 importance of the jaw, the extreme repetitive tensile and shear stresses associated with mastication, and the frequent use of radiotherapy.”

Segmental defects are also created during reconstruction surgery after trauma, not only after cancer. Please include this information unless it is important to draw attention to the cancer treatment.

Response: We intentionally highlighted the challenges that are specific to cancer and the mandible. For cancer, these include the need for radiotherapy as part of the treatment and patients need to undergo repeated surveillance scans using either computed tomography (CT) or magnetic resonance imaging (MRI) where metallic plates create substantial artifacts. The problems caused by metallic plates extend beyond imaging artifacts which may delay the detection of cancer to include unwanted alterations in the radiotherapy dose distributions which may increase the likelihood of the cancer recurring and the rate of

radionecrosis due to localised increases in radiotherapy dose. Whilst trauma may also require surveillance imaging, the number of repeated scans is generally less, and the issue of radiotherapy is absent.

To ensure that this intention is clear, we have changed the sentence to:

(P4, Lines 113 - 120) “Whilst critical-sized bone defects can occur anywhere in the skeleton and have a variety of causes, segmental defects of the mandible created in the treatment of cancer are one of the most challenging to manage due the high aesthetic and functional importance of the jaw, the extreme repetitive tensile and shear stresses associated with mastication, the frequent use of radiotherapy, and the need for prolonged surveillance imaging. Solving this challenge in the ‘worst case’ will have widespread implications for the use of SGBR in other locations and etiologies.”

7. Line 102 – Please also add other Ti alloys (Ti64) and other metallic materials used clinically (SS316L).

Response: This has been changed to:

(P4, Lines 121 - 124) “Large segmental defects are traditionally reconstructed using vascularised autologous bone grafts taken from sites such as the fibula, pelvis, or scapula and fixated using plates made of titanium (Ti), Ti alloys (e.g., Ti-6Al-4V), and other metallic materials such as austenitic grade stainless steel (e.g., SS316L).”

8. Line 147 – “...however, it has never been used for segmental bone defect repair.” Has PEK been used for other implantable medical devices? Please include references.

Response: To the best of our knowledge and following a thorough literature search, there is no documented clinical application of laser-sintered PEK (polyether ketone) as an implantable medical device in humans, and it is not currently FDA, EMA, or TGA approved. We have used laser-sintered PEK in the preclinical ovine studies described in Figure 2 of our manuscript, including applications such as ‘sawtooth’ cylinders, bioreactors, and non-segmental bone defects.

Given that PEK was specifically developed for laser sintering using the EOSINT P800 printer, it is unlikely to have been employed with other techniques such as fused filament fabrication or milling. We acknowledge, however, the possibility that relevant literature

might be published under different naming conventions, particularly since the commercial name of PEK is EOS HP3 PEEK. Thus, PEK may be grouped under the broader PAEK family and confused with PEEK (polyether ether ketone) or PEKK (polyether ketone ketone). This is illustrated by the *in vitro* study by Cvrček L et al. *Nanostructured TiNb coating improves the bioactivity of 3D printed PEEK*, published in *Materials & Design*, Volume 224, 2022, <https://doi.org/10.1016/j.matdes.2022.111312>.

To ensure thoroughness, we repeated our literature search using the terms “PEK,” “HP3,” “PEEK,” and “PEKK.” Despite this, we found no documented evidence supporting the use of PEK as an implantable medical device or in segmental bone defect repair in humans or in any animal models (large or small).

We identified a publication by Yu et al. (2018) that describes the use of PEK/biphasic bioceramic composites in a rabbit jawbone defect model. However, their methods reference a previous study by the same group (Leng et al., 2015), which clearly indicates that the material used was PEEK, not PEK.

- Yu, H.D., et al. (2018). *Exogenous VEGF introduced by bioceramic composite materials promotes the restoration of bone defect in rabbits*. *Biomed. Pharmacother.* 98, 325–332.
- Leng, W.D., et al. (2015). *Preparation and mechanical properties of polyetheretherketone/odontogenic biphasic bioceramic composites*. *Chin. J. Tissue Eng. Res.* 19, 8418–8422.

Additionally, we found a few instances where PEK was used in non-bone-related applications. For example, PEK has been used as perivascular cuffs in small animal models:

- Cheng, C., et al. (2005). *Shear stress affects the intracellular distribution of eNOS: direct demonstration by a novel in vivo technique*. *Blood* 106, 3691–3698.
- Mohri, Z., et al. (2014). *Elevated Uptake of Plasma Macromolecules by Regions of Arterial Wall Predisposed to Plaque Instability in a Mouse Model*. *PLoS ONE* 9(12): e115728. doi:10.1371/journal.pone.0115728.

To our knowledge, these limited examples represent the only reported uses of PEK *in vivo* and none involve segmental bone defect repair.

9. Line 148 – “PEK can be selectively laser-sintered (SLS)”

Please use the correct terminology for the AM technology according to ISO/ASTM 52900-15. Powder Bed Fusion using Laser Sintering (PBF-LS). Homogenize for the entire manuscript.

Response: We have included the following:

(P5, Lines 179 - 182) “PEK can be additively manufactured through the process of powder bed fusion using laser sintering (PBF-LS), which is commonly known as selective laser sintering (SLS), and annealed for improved mechanical performance with a micro-rough surface topology that promotes tissue integration.”

In addition, we have revised our terminology to laser sintered PEK (LS-PEK) throughout the manuscript for brevity. By way of explanation, LS is used by other publications in Nature Communications (e.g., Krumins, E., Crawford, L.A., Rogers, D.M. et al. A facile one step route that introduces functionality to polymer powders for laser sintering. Nat Commun 15, 3137 (2024). <https://doi.org/10.1038/s41467-024-47376-4>). Furthermore, we are concerned that some readers may not understand that PBF-LS is the same process that other studies refer to as LS or SLS. If the reviewer feels that this is unacceptable, then we will change our revised terminology to PBF-LS-PEK.

10. Line 171 – From line 171 through the end of the introduction section, it looks more like an abstract than an introduction section. Please reduce the information. Just state what the approach you followed was and state the main result in one sentence.

Response: We have reduced the text to:

(P5, Lines 198 - 200) “Remarkably, all sheep demonstrated normal masticatory function due to osseointegration of the PIII-treated LS-PEK implant with progressive stress-driven osteoconduction through the scaffold. “

11. Please include references to other stiffness-matched approaches (more elastic materials and porous implants) for metallic implants and what problems this approach tackles.

For example: Khattab, N.R., Olivas-Alanis, L.H., Chmielewska-Wysocka, A. et al. Evaluation of stiffness-matched, 3D-printed, NiTi mandibular graft fixation in an ovine model. *BioMed Eng OnLine* 23, 105 (2024). <https://doi.org/10.1186/s12938-024-01289-x>

Response: This reference and other relevant publications have been added, and the statement has been modified to:

(P4, Lines 133 - 141) “The stress shielding associated with high-modulus materials may be lessened by ‘stiffness matching’ which can be achieved by changing material composition through the use of composites or by changing its geometry, i.e., location, shape or porosity. Unfortunately, this is not always feasible due to anatomical constraints and whilst some problems may be overcome by optimizing the mechanical properties, this does not address issues related to X-ray perturbation during postoperative radiotherapy and associated imaging artifacts. X-ray perturbation can increase the dose of radiotherapy to healthy bone, under-dose the target volume, and interfere with surveillance imaging modalities, impeding early detection of infection and local recurrence of cancer (Fig. 1).”

12. Results: Line 194– “replicate selective laser melted (SLM) grade 23”

Please use the correct terminology for the AM technology according to ISO/ASTM 52900-15. Powder Bed Fusion using Laser Beam (PBF-LB). Homogenize for the entire manuscript.

Response: We have revised text to:

(P6, Lines 213 - 216) “For the first study (phase 1), the bone-implant interface of PIII-treated LS-PEK sawtooth cylinders was compared with replicate grade 23 Ti cylinders manufactured by powder bed fusion using laser beam (PBF-LB) that were implanted into the mandible for 8 to 12 weeks.”

13. Line 268 – Figure 3.A

Indicate what views of the implant are shown in the image (top, bottom, cross section...) Also, show the structure used for the TCP structure.

Response: We have revised the figure as suggested, and Figure 3.A has changed to Figure 4A.

14. Line 268 – Figure 3.B

Why does the implant shown in the mechanical setup have an incomplete porous structure?

Why is the model restrained at the front teeth? The boundary conditions seem different for this setup than for the implanted bones.

Response: Note, Figure 3.B has been changed to Figure 4B. Multiple mechanical tests using many different set-ups have been performed on multiple implants fixated to sheep cadavers to validate the FE analyses in the design optimisation process. In Figure 4B, the sheep mandible was thinner than other sheep and thus the design, which had to also accommodate the β TCP lattice, resulted in an incomplete gyroid structure at the posterior aspect of the mandible. This was not the case for implanted sheep, where we were able to avoid this issue through further design modifications. We used this figure because we believe it demonstrated one mechanical setup well. However, we have changed the figure and included an example where the gyroid scaffold was complete. This mechanical test setup was one of many designed and used to replicate the implanted scaffolds. In reality, although the maximum muscle force can be estimated from physiological cross-sectional areas, these have not been validated and the actual (true) mechanical properties in the sheep are unknown. This is made more difficult because it is unknown how functional the detached musculature will be and what force sheep actually apply during mastication. Furthermore, sheep have a cyclical chewing action which is challenging to simulate in unidirectional mechanical tests. In the figure, the condyles are not fixed and muscle force from the pterygomasseteric sling and temporalis muscle are generated by the high-tensile bands placed around and secured to the mandible. The incisor teeth are constrained to represent the longest cantilever applied to the implant, representing a worst-case scenario. However, multiple other mechanical test set ups were applied to represent molar tooth mastication with both vertical, oblique, and horizontal loads applied.

15. Line 268 – Figure 3.C

Why is there a gap between the porous gyroid region and the solid “shell”? Figure 3A does not show this gap, and it shows that the solid and porous parts are joined.

Response: Note, Figure 3C and 3A have been updated to Figure 4C and 4A. The gap is due to the β TCP lattice which sits within the LS-PEK frame adjacent to the gyroid scaffold and an artifact of the slightly oblique angle that was chosen to demonstrate this potential space. The LS-PEK gyroid scaffold contacts the bone at each end and the β TCP lattice also contacts the bone at each end. In the region where the β TCP contacts the bone, there is a space in the LS-PEK scaffold. There is also a 1 mm gap between the LS-PEK frame and the β TCP lattice to allow for installation.

16. Line 304 – *“ μ CT demonstrated osteogenesis through the gyroid structure ranging from 1.16 cm³ to 2.65 cm³ in some cases, bridging the entire bone defect.”*
Would it be useful to relate these numbers to the porous volume of the implant? What’s the meaning of this 1.16-2.65 cm³ volume compared to the defect?

Response: The fraction of the total void volume replaced by bone has been added to the manuscript as per below:

(P11, Lines 329 - 331) “ μ CT demonstrated osteogenesis through the gyroid structure ranging from 0.93 cm³ to 2.71 cm³, representing 11% to 38% of the available LS-PEK scaffold volume. In some cases, new bone bridged the entire 6 cm bone defect (Fig. 6).”

17. Line 362 – *“The shift of peak VM stress to the mandibular bone reduces the risk of 363 ductile-related damage, such as yielding and fatigue failure.*

Please rewrite for better understanding.

Response: We have revised the sentence to:

(P15, Lines 402 - 411) “In contrast, a more even stress distribution was seen in the middle of the LS-PEK frame with osseointegration, and more VM stress (65 MPa) was transferred to the posterior mandible, indicating the bone actively shared the occlusal load (Fig. 7Fii). The shift of peak von Mises stress from the implant to the mandibular bone suggests that the implant is bearing less load, thereby reducing the risk of ductile-related damage such as yielding or fatigue failure in the implant. In the model without osseointegration, the region with the highest tensile MPS (95 MPa) was around the screw holes (Fig. 7Gi). However, when osseointegration and bone growth occurred, the load was transferred to the LS-PEK frame through the bone-implant interface, rather than relying on the screws (Fig. 7Gii). Although some concentration of MPS was observed in the left upper re-entrant corner of

the scaffold, the majority of the MPS was distributed on the upper surface of the posterior mandible.”

18. Line 388 – Figure 6

What is the meaning of the cracks on the host mandible? For me, they were promoted by how the mechanical testing setup was designed. It will be more useful to have a crack initiator site or grow information for the construct or the host bone (away from the resin fixation areas). Like the information presented in Figure 3.B

Response: Note, Figure 6 and Figure 3B has been updated to Figure 7 and Figure 4B. The reviewer is correct in noting that the observed crack patterns in the host mandible are influenced by the mechanical testing setup. This FE simulation was designed to replicate *ex vivo* experimental conditions rather than physiological loading scenarios. In these mechanical tests using the explanted LS-PEK scaffold and mandible, the primary objective was to assess bone integration of the scaffold at the bone–implant interface under controlled loading and the secondary objective was to assess how osteoconduction affected the mechanical properties under load. Thus, we designed the set up to be a hybrid mechanical test which assessed both tensile strength and bending moment.

Importantly, the results demonstrated the level of osseointegration was sufficiently high for failure to occur in the native bone, but not in the implant or at the implant–bone interface. This outcome suggests robust mechanical integration between the implant and host bone, which aligns with the intended purpose of evaluating the effectiveness of the construct in supporting bone ingrowth and integration.

19. Line 389 – Caption Figure 6

Please use F(i), F(ii), G(i), G(ii) in the caption, or change the figure to f1,f2,g1,g2.

Response: This has been corrected. Note, Figure 6 has been updated to Figure 7.

20. Methods: Line 503 – “The angle of the mandible was selected because it is a site of maximum stress”, Under what loading? Unilateral? Frontal? Please include references.

Response: Orassi et al (ref 40) have compared the estimate biomechanics of humans and sheep to validate the sheep model for plate fixation. In both cases the estimated maximum

principal strains were highest around the angle of the mandible with tensile and compressive strains up to 1,200 $\mu\epsilon$ and -1,750 $\mu\epsilon$ during intercuspal (bilateral) and unilateral clenching.

We have revised the text to:

(P20, Lines 578 - 580) "The angle of the mandible was selected because it is a site of maximum stress and strain during intercuspal (bilateral) and unilateral clenching and has similar anatomical features to humans, particularly the ratio of cortical to cancellous bone."

Reference: Orassi V, Duda GN, Heiland M, Fischer H, Rendenbach C, Checa S. Biomechanical Assessment of the Validity of Sheep as a Preclinical Model for Testing Mandibular Fracture Fixation Devices. *Front Bioeng Biotechnol.* 2021 May 6;9:672176. doi: 10.3389/fbioe.2021.672176. PMID: 34026745; PMCID: PMC8134672.

21. Line 523 – *"The 523 scaffold's pore structure is composed of 1 mm pores arranged in a 3D array of periodic unit cells 524 based on Schwarz P-surfaces."*
What's the strut size? What's the resulting porosity? How did you measure/define the strut and pore size?

Response: Both the pore and strut size for the Schwarz P TPMS employed is 1 mm. The strut size is defined as the minimum diameter/thickness of the cross section of a strut while the pore size was defined as the maximum diameter sphere that could fit through the pores of the lattice (see Liu et al below). The resulting porosity was 50% as determined by volume fraction.

We have revised the text to:

(P20, Lines 596 – 599) "The scaffold's structure is a 3D array of periodic unit cells based on Schwarz P-surfaces with a pore size of 1 mm and strut size of 1 mm. The resulting porosity was 50% as determined by volume fraction. Schwarz P is a class of TPMS optimised for tissue integration and vascularization."

Reference:

Liu, H., Liu, L.-L., Tan, J.-H., Yan, Y.-G. and Xue, J.-B. (2023), Definition of Pore Size in 3D-Printed Porous Implants: A Review. *CBEN*, 10: 167-173.

<https://doi.org/10.1002/cben.202200043>

22. Line 533 – *“The SLS-PEK frame was custom designed to fit each sheep’s left mandible using a clinically validated VSP process where the ablation and reconstruction are digitally simulated.”*

Please add a reference for the clinically validated VSP process.

Response: The following reference has been added (**reference 5 in manuscript**).

Leinkram, D. *et al.* Occlusal-based planning for dental rehabilitation following segmental resection of the mandible and maxilla. *ANZ J Surg* **91**, 451–452 (2021).

23. Line 548 – *“The middle section bridges the bone defect employing a single gyroid TPMS scaffold architecture (3.5 mm unit size and 1.6 mm porosity) with solid reinforcement superiorly and inferiorly.”*

What’s the porosity level (%)? Why was it changed from 1mm to 1.6 mm? Please replace “1.6mm porosity” with “1.6 pore size”. If you haven’t done it yet, define how you measured the pore size. What about the strut size? How were these parameters selected? It is stated that it is based on numerical modelling, but were other sizes evaluated? Please include references if useful.

Response:

(1) “1.6 mm porosity” has been changed to 1.6 mm pore size.

(2) The following definitions have been added:

(P21, Lines 623 – 625) “The strut size is defined as the minimum diameter/thickness of the cross section of a strut while the pore size was defined as the maximum diameter sphere that could fit through the pores of the lattice.”

This definition of pore size is now consistent with the definition of pore size and strut size used for the Schwarz P TPMS.

(3) The change from 1 mm used for the Schwarz P to 1.6 mm for the gyroid scaffold was due to strength and cleaning constraints (see below).

(4) The following explanation was added:

(P21, Lines 625 – 633) “A 50% volume fraction porosity level was chosen to give equal weighting to the strength of the struts and the cleanability of the residual

powder after printing. A 3.5 mm cell size was chosen after a series of test lattices of varying sizing from 5 mm cell sizes down to 2 mm cell sizes were designed, printed, and cleaned. It was observed that cell sizes below 3.5 mm were difficult to clean effectively and cell sizes below 3 mm had struts break during the cleaning process. Gyroid replaced the Schwartz P structure used in phase 2 studies due to its isotropic mechanical properties and better termination at the surface of the implant. Gyroid structures are also stronger and promote better osteoconduction.”

24. Line 554 – “The lattice cell size was selected to be 1.875 mm, with a beam thickness of 0.4 mm”

Define what lattice cell was used. And how did you measure the pore and strut size? (if different than TPMS Gyroid). Was it the one in line 563? If so, please rewrite to include that information in line 554.

Response: The lattice cell is a Body-Centered Cubic lattice, which as the reviewer indicated was described in line 563.

We have revised the text to:

(P21, Lines 637 – 641) “The cell size of the Body-Centered Cubic β TCP lattice was selected to be 1.875 mm, with a beam thickness of 0.4 mm; these specifications were within the limits of the 3D printer used (Lithoz Cerafab 7500), allowed for effective post-processing (removal of residual ceramic slurry), provided adequate structural integrity, and permitted the lattice space to be backfilled with GelMA hydrogel.”

25. Line 558 – “*The design process used specialized lattice generation software to create 559 a lattice structure within the confines of the desired geometry to ensure that the lattice would 560 align with the boundaries of the planned insertion void.*”

What software was used for the generation of the porous structure?

Response: The software used was nTopology,

We have revised the text to:

(P21, Lines 641 – 644) “The design process used nTopology, a specialised lattice generation software, to create a lattice structure within the confines of the desired geometry to ensure that the lattice would align with the boundaries of the planned insertion void.”

26. Line 575 – *What are “2,ss elements”? Please correct, rewrite, or define.*

Response: Thank you for identifying this mistake. We have revised the text to:

(P21, Lines 657 – 660) “After surface smoothing and refinement, the 3D solid models of mandible, scaffold, and screws were meshed using 4-node tetrahedral elements in ScanIP, containing 3,826,530 elements with a total of 2,092,518 degrees of freedom (DOFs).”

27. Line 587– *“Two osseointegration conditions were considered: the first scenario assumed a lack of osseointegration, while the second scenario mimicked the presence of osseointegration between the scaffold and the host mandible.”*

Is the second condition assuming bone ingrowth? Or it is only considering a tied bone between the host bone and the porous implant.

Response: We appreciate the reviewer’s comment. In our current FE simulations, osseointegration was represented by modifying the interaction properties at the bone-implant interface rather than explicitly modelling bone ingrowth. While μ CT and histological evidence confirm the presence of new bone tissue within the porous scaffold, this newly formed bone is sometimes immature and may not always contribute to load bearing during the early stages of healing. To simplify the model and isolate the mechanical influence of the interface condition, we focused on two representative scenarios: a fully bonded (tied) interface to mimic idealized osseointegration, and a frictional/sliding contact to represent the absence of integration. This approach allows us to systematically assess how interface bonding influences structural stability while avoiding assumptions about the evolving mechanical properties of immature bone tissue. We believe this simplification is appropriate for capturing the early mechanical environment and provides useful insights into the role of the bone–implant interface in supporting load transfer.

28. Line 614 – *“The load and boundary conditions were set to simulate the biting forces for the sheep mandible.”*

What are those boundary conditions?

Why wasn’t it considered a chewing model for the evaluation of the construct?

Response: The boundary condition was applied to the condyle region by fully constraining all degrees of freedom to replicate the in-vitro experimental setup.

Regarding the use of a chewing model, in the current study, we employed a simplified biting force model rather than a full chewing simulation to evaluate the structural integrity of the construct. This decision was made based on several considerations: (1) The biting force model provides a controlled and repeatable loading condition that enables a clearer assessment of structural response and stress distribution within the implant and surrounding bone. It is particularly suitable for evaluating peak loading conditions, which are critical for understanding the construct's load-bearing capacity and potential failure modes. (2) Incorporating a full chewing cycle introduces complex, time-dependent, and multidirectional loading patterns. While such a model would provide more physiological relevance, it also significantly increases the computational complexity and introduces uncertainties due to the high variability in chewing patterns among individuals. These factors can make it challenging to isolate and interpret the mechanical performance of the implant itself.

Therefore, the biting force model was selected as a first step to investigate the mechanical integrity and stability of the reconstruction under representative maximum loading. We agree that simulating chewing behaviour would provide valuable insights, particularly for long-term fatigue behaviour, however, we need to resolve the uncertainties to ensure that the model is clinically appropriate before pursuing this direction for future studies.

29. Line 637 - Was there any morphological assessment of the implants after printing (SLS and DLP) or postprocessing? How different are they from design?

Response: Morphological assessment of each LS-PEK implant was performed in two ways: (1) Point to point measurements were performed on the critical dimensions of the LS-PEK frames including screw hole spacings, screw hole diameters, overall length, width, flange thickness, and width of β TCP lattice receptacle. These measurements were performed with Vernier callipers and compared to the virtual design file via Materialise Magics software. A printing orientation was chosen to reduce the severity of dimensional inaccuracies for critical dimensions. The most severely affected critical dimension was the flange thickness which experienced a 0.3 mm increase in flange thickness that is typical of LS additive manufacturing for so called 'downskin' surfaces. All other dimensions were within ± 0.2 mm. The increase in flange thickness was not seen as critical dimensional

inaccuracy as the concern was potential weakening of the flange caused by a reduction in flange thickness. (2) μ CT was performed to check the internal geometry and to ensure that all of the unsintered powder had been removed during postprocessing. This demonstrated a high degree of conformance with the design specifications and minimal residual powder; however, formal measurements were not obtained due to the complex structure.

Morphological assessment of the β TCP lattice was more limited due to its fragile structure, which easily cracked on manipulation. The overall dimensions were confirmed at the time of implantation by demonstrating that the lattice fitted within the LS-PEK frame, which had a 1 mm tolerance. The internal structure was confirmed by testing non-implanted samples using μ CT. This was done to determine a method for calculating bone within the β TCP, which was problematic due to its high density.

30. Line 820 – *“The volume of new bone formation at the defect site was quantified using the segmentation statistics feature of 3D Slicer.”*

How do you confirm that you are comparing the same volume/area? Are there any boundaries?

Response: We have refined our methodology in the revised manuscript to ensure a more accurate and consistent quantification of new bone formation within the defect site, specifically in the PEK and β TCP regions. In the prior version, segmentation was conducted on reconstructed NIfTI image files using 3D Slicer®, where the region of interest (ROI) was defined using the “volume rendering” and “crop volume” tools. To standardize the boundary between native bone and new bone, we used the first row of screw holes adjacent to the osteotomy plane as fixed anatomical landmarks. A 5 mm offset was applied from the plane formed by these screw holes (both anteriorly and superiorly) toward the defect to define the starting boundary of new bone formation.

In the updated method (**P27, Lines 911 – 932**), we took additional measures to ensure consistency in volume comparison and to clearly delineate boundaries. While the same anatomical references (i.e., screw hole positions) were used to define the osteotomy plane, we also aligned μ CT scans with histological sections to validate the segmentation of bone, β TCP, and LS-PEK regions. This alignment helped refine the thresholding process and minimize inclusion errors due to artifacts such as partial volume averaging, beam hardening, and scatter effects. A broader CT value threshold range (1800–5300 HU) was applied to segment newly formed bone, while a threshold of 5300 HU and above was used

to segment β TCP. To address the issue of artifact overlap at the interface of bone and β TCP, Boolean subtraction was performed to remove erroneously included voxels from the bone segmentation. A geometric offset of +200 μ m was also applied to compensate for the underestimation of β TCP strut thickness during segmentation. Finally, the remaining segmentations were cross-referenced with the digital 3D implant models and the baseline CBCT scans taken immediately after surgery to verify the original boundary between the native bone and the defect site. This ensured that only the bone formed within the predefined defect region was quantified. The new bone volume was further subdivided into regions within the β TCP and LS-PEK lattices using Boolean operations with non-latticed implant models and native bone segmentations. All volumetric measurements were performed using the segmentation statistics feature of 3D Slicer®, ensuring consistency across all samples. The entire workflow and boundary definitions are illustrated in **Supplementary Figures S11a and S11b** and in **Figure 6**.

31. Line 826 – *“Mechanical testing of the full mandible used two high-tensile straps designed to simulate the pull of the medial pterygoid and masseter muscles whilst loading the condyle and incisor regions and allowing rotation of the condyle/coronoid.” How is the stiffness of the straps related to the muscles' properties?*

Response: We thank the reviewer for another insightful question. The high-tensile straps used in the mechanical testing setup were not intended to replicate the mechanical properties of the medial pterygoid and masseter muscles. Instead, they were strategically positioned to align with the anatomical orientation and attachment sites of these muscles. The primary goal of the straps was to apply loading in a physiologically relevant direction while avoiding rigid constraints that could artificially restrict natural mandibular deformation. This setup enabled controlled force application that mimics the general line of action of the masticatory muscles, thus preserving the mandible's structural response under functional-like conditions.

While the straps do not capture the complex viscoelastic or contractile behaviour of actual muscle tissue, their flexibility provides a simplified yet effective means of reproducing the directional influence of muscular loading during mastication-related testing.

In closing, we thank you once again for your thoughtful review and constructive feedback, which have helped us to improve the quality and clarity of our manuscript.

14th August 2025

Dear Reviewers,

We sincerely thank both reviewers for their time and valuable feedback, which has helped us improve the quality of our manuscript. We are especially grateful to Reviewer 2 for accepting the manuscript for publication and to Reviewer 1 for the constructive comments and suggestions. Below we provide a point-by-point response to the comments listed in italics.

Reviewer #1:

The reviewer judged that most of the comments previously provided by the reviewer were accepted by the authors and revised accordingly. However, the reviewer recommends that the authors make further revisions to address the remaining shortcomings in the manuscript. For this manuscript to be improved, the reviewer would like to make comments as follows:

- 1. All graphs should include p-values. As there are still figures in which p-values are missing, the reviewer recommends that the authors carefully review all graphs and ensure the inclusion of p-values to enhance the reliability and credibility of the results.*

Response: P values from statistical analyses are now included in all graphs that compare data between samples. We have, therefore, added p-values to the following:

- **Figure 2Aiv**
- **Figure 2Biii**
- **Figure 2Biv**
- **Figure 4H**
- **Figure 5R**
- **Figure 5S**
- **Figure 5T**
- **Figure 7C**
- **Figure 8**

We have not added p values to the following figures:

- **Figure 4B:** This is a single force displacement curve used as an example of in vitro mechanical testing done for each scaffold design, and so no comparisons were made.
- **Figures 5O-Q:** These figures show individual sheep data corresponding to the aggregated data with p values shown in **Figures R-T**.

2. *The manuscript still contains a substantial amount of simulation data, and it remains difficult to clearly distinguish between the native bone and the scaffold following implantation. Therefore, the reviewer recommends that the authors explicitly indicate, for each figure, which colors correspond to bone and scaffold, and use arrows within the images to clearly highlight the location of newly formed bone.*

Response: The importance of the simulation data for validating the mechanical drivers of osteoconduction is addressed in point 3 below. With regards to the figures, please note that the LS-PEK scaffold is radiolucent, so it is not visible on any of the cone beam CT or μ CT images. We have added the following to address the reviewer's recommendation:

- **Figure 2Bii** – Arrows to indicate newly formed bone and description in the legend regarding what colour corresponds to bone.
 - **Figure 4E** - A description in the legend regarding which colours correspond to new bone.
 - **Figure 4F** - A description in the legend regarding which colours correspond to new bone.
 - **Figure 4G** - A description in the legend regarding which colours correspond to new bone.
 - **Figure 5J** - Arrows to indicate newly formed bone and description in the legend regarding what colour corresponds to bone.
 - **Figure 5K** - Arrows to indicate newly formed bone and description in the legend regarding what colour corresponds to bone.
 - **Figure 5L** - Arrows to indicate newly formed bone and description in the legend regarding what colour corresponds to bone.
 - **Figure 5M** - Arrows to indicate newly formed bone and description in the legend regarding what colour corresponds to bone.
 - **Figure 5N** - Arrows to indicate newly formed bone and description in the legend regarding what colour corresponds to bone.
 - **Figure 5U** - Arrows to indicate newly formed bone versus the β TCP scaffold.
 - **Figure 5V** - Arrows to indicate newly formed bone versus the β TCP scaffold.
 - **Figure 5W** - Arrows to indicate newly formed bone.
 - **Figure 6B** - A description in the legend regarding which colours correspond to new bone.
 - **Figure 8** - A description in the legend regarding which colours correspond to new bone and arrows to indicate newly formed bone.
3. *Because sheep exhibit highly active autologous bone regeneration, the most critical factor in validating the efficacy of the scaffold presented in this study is to clearly demonstrate whether the implanted cells have successfully differentiated into bone cells. To this end, the authors should employ specific markers that distinguish implanted cells from host cells to directly verify that the implanted cells have proliferated within the scaffold and effectively undergone osteogenic differentiation. Particularly in this*

study, where the use of a control group with segmental mandibular defects without scaffold implantation is not feasible due to ethical constraints, such analysis is considered essential to assess the contribution of the implanted cells. Considering the standard expected for a high-impact journal like Nature Communications, stating that these experiments will be addressed in future work appears insufficient to substantiate the scaffold's functional efficacy. Therefore, the reviewer strongly recommends that the authors perform the necessary experiments and present both quantitative and qualitative data to support their claims. In addition, the reviewer recommends that the authors explicitly write that, for ethical reasons, a control group with unreconstructed segmental mandibular defects could not be included in the study.

Response: Before addressing the reviewers main concern (see below), we have added to the methodology that for ethical reasons a control group with unreconstructed segmental defects could not be included in the study as requested.

Page 21, Line 605-606 “For ethical reasons, a control group with unreconstructed segmental defects could not be included in the study.”

To address the request to *employ specific markers that distinguish implanted cells from host cells to directly verify that the implanted cells have proliferated within the scaffold and effectively undergone osteogenic differentiation*, please allow us to explain why this task (to our knowledge) is near impossible in a defect size of this magnitude which contains a hard implant:

For *in vivo* tracking, fluorescently tagged cells are routine in mouse tumour models owing to the mouse being able to be inserted into imaging equipment (such as *in vivo* μ CT). In contrast, a sheep cannot be imaged this way and tissue penetration is insufficient to allow standard CT imaging approaches. For *ex vivo* tracking on animal sacrifice, standard paraffin serial sectioning cannot be used for hard scaffold implants as their density exceeds that of the paraffin wax in which they are embedded. Whilst bone can be decalcified to make it soft enough to embed in paraffin, one cannot decalcify a polymeric implant. Instead, we have used resin embedding and ground sectioning where the stiffness of the plastic MMA resin is similar to the bone and the implant and a precision bandsaw with diamond-coated blade can be used to pass through it. The resulting slice of tissue is pre-mounted onto a large slide and ground and polished down to a near cell-like thickness in order to probe with stains and antibodies. This method generates superior slices of large and hard implants and gives holistic images of the entire defect site in one plane – but it cannot be used to create serial sections like paraffin sectioning as the tissue has to be “ground and polished away” to make it perfectly flat for staining and imaging. Unfortunately, at least 95% of the tissue is lost in the process of achieving one perfect whole defect slice. This makes the likelihood of identifying fluorescently labelled implanted cells statistically improbable. In addition, the process of embedding the explanted scaffold and host tissue in resin takes many weeks, during which the cells are constantly exposed to extremely robust solvents (such as alcohol, xylene and resin solutions), not just for a couple of hours like small samples in paraffin – but for weeks. Thus, even if the

cells retained their fluorescent tag (or indeed any tag) during the implantation period, the likelihood of that tag being unaffected by the extreme processing conditions required for resin embedding is very low. This is why, to our knowledge, there are no published work or protocols using reporter cell lines in a defect of this magnitude with a hard implant in a sheep model.

Practicalities aside, we have attempted to provide evidence in the manuscript that the ADSCs are contributing to bone formation in the scaffold through supporting preliminary *ex vivo* data and by providing greater emphasis in the paper regarding the location of the new bone and its proximity to the area that the cells were introduced. Three years of extensive *in vivo* experiments comparing different biomaterials and cellular constituents are summarised in **Figure 2**. This preliminary work demonstrated that the combination of osteogenically differentiated ADSCs within GelMA infused in a β TCP scaffold yielded increased *in vivo* heterotopic osteogenesis within SL-PEK bioreactors compared to β TCP \pm GelMA. **Figure 2C** shows histology and immunofluorescent markers of osteogenesis confirming that the osteogenically differentiated ADSCs had differentiated into bone-lineage cells using the ADSC differentiation protocol described in the manuscript, which we are confident has not occurred through osteoconduction due to their heterotopic location. (Legend: Immunohistochemistry confirming viable osteogenic stem cells (GelMA autofluorescence: green, bone lineage CD44 expressing cells [osteoblasts, osteoclasts and/or osteocytes]: red, and DAPI stained cell nuclei: blue). These earlier experiments support the histological analysis of the explanted segmental mandibulectomy samples shown in **Figure 6Av** demonstrating bone cells within the centre of the β TCP scaffold, remote to the areas of osteoconduction. Taken together, these data provide evidence that the ADSCs differentiated into bone lineage cells and remained viable within the β TCP scaffold up until the point of explantation.

We also do not want to overstate the effect of the ADSC's, as the innovation in this paper is not the use of ADSC's – it is the success of the model itself. The implanted cells are playing a minor role in the success of the scaffold. We believe that clinical success of the implant was primarily due to osseointegration with native bone. The experimental data shows that most new bone is not from the implanted cells and has grown from the cut edge of the native bone through the process of osteoconduction. This is likely primarily driven by mechanical stimuli due to the material properties of the scaffold, verified by the biomechanical simulations, which is why we have given the simulated data primacy. In contrast, the process of osteoinduction involving the autologous cell-laden GelMA, whilst desirable, is deemed to be of lessor importance. Reiterating from the manuscript, the ADSC's were principally included to increase the bioavailability of the β TCP rather than directly provide new bone involving, for example, ADSC-derived osteoblasts. Regardless of their role, it ultimately proved immaterial given that osteoinduction was non-essential. It is, therefore, difficult to justify exposing further sheep to experimentation, (which is extremely resource intensive and costly) solely for the purpose of further verifying the state of the implanted cells. Verification would require using stable constitutive or inductive fluorescent reported ADSC lines, with the latter potentially enabling osteoblast and osteocyte lineage tracing. This is a future endeavour that we would like to work on as the reviewer suggestion is

indeed a good one, it's just never been done before and may hold practical considerations that render it impossible.

In summary, whilst we agree with the reviewer on most aspects, we disagree that the most critical factor *in validating the efficacy of the scaffold is to demonstrate whether the implanted cells have differentiated into bone cells*. We contend that the most critical factor in validating the efficacy of the scaffold is clinical success as demonstrated by the wellbeing of the sheep, i.e., wound healing, masticatory efficiency, weight gain, absence of pain/distress, and durability. The role of additional analyses (including imaging, simulation data, mechanical testing, macroscopic evaluation, and microscopic analyses) is to better understand the mechanisms for success or failure. We would like to highlight that although immature *sheep exhibit active autologous bone regeneration*, we specifically chose mature sheep because (in contrast to immature sheep) bone turnover in mature sheep is similar to adult humans making them an appropriate experimental model. Further their extreme masticatory requirements make successful reconstruction even more difficult than in humans. Consequently, reconstruction of a clinically relevant (6 cm) segmental mandibulectomy defect without metal plate fixation has never been published before. Thus, the findings are highly novel and important, regardless of mechanistic considerations. Lastly, to reiterate, the practicalities in achieving evidence of the labelled cells causing osteogenesis is currently not possible in a defect of this size, with hard implants and with the imaging modalities available (size restricted) and the necessity to use resin embedding and ground sectioning thus losing most of the tissue – such approaches are demonstrated in mice with soft tissue tumours which can be paraffin embedded and serial sectioned, but not for sheep with hard implants.

4. *With regard to the fabrication of scaffolds incorporating functional vascular and musculoskeletal structures for bone defect repair, the reviewer finds that the introduction section still lacks sufficient discussion of existing biofabrication techniques and alternative biomaterials. Therefore, the reviewer recommends that the authors expand the introduction to include a more comprehensive overview of relevant biofabrication methods and biomaterials. Additionally, the authors should clearly explain why the specific SLS printing technique and biomaterial used in this study are superior compared to these alternatives, thereby emphasizing the novelty and significance of their approach.*

Response: We have further revised the introduction to include a more comprehensive overview of relevant biofabrication methods and biomaterials as follows:

Page 4 In 141- 224

“There is considerable interest in using SGBR in the form of patient-matched medical devices (PMMDs) that are additively manufactured to precisely match the defect size and shape and abrogate the need for autologous bone grafts. Despite this, SGBR has failed to deliver constructs that are suitable for use

in clinical practice because most tissue engineering strategies employ bioresorbable scaffolds that are progressively replaced by bone as they degrade.^{11,18-21} Examples of biomaterials trialed for reconstructing segmental bone defects in large animal models include ceramics such as hydroxyapatite and beta-tricalcium phosphate (β TCP) and various polymers (poly-caprolactone, poly-lactic acid, poly-lactic-co-glycolic acid, polyurethane and hydrogels), in combination with growth factors and cellular components. This concept is attractive; however, it relies (unrealistically) on synchronizing the rates of scaffold degradation and osteogenesis to achieve the desired biomechanical properties in the medium- to long-term. Furthermore, whilst most bioresorbable scaffolds are designed to withstand compressive loads, they require reinforcement with metal plates or intramedullary nails to address issues with fixation, bending moment, and torsional stability. In fact, there are neither preclinical large-animal studies nor clinical studies published where SGBR has been used to reconstruct segmental bone defects without metal plate augmentation, and most studies reconstruct bone defects that are too small to be relevant to oncology, which are typically 6 cm or more. The most promising 3D-printed biomaterial for SGBR thus far appears to be a combination of PCL and β TCP, which can be fabricated as a porous biocompatible construct with suitable structural and osteoconductive properties. However, even with rigid fixation, all bioresorbable scaffolds will ultimately fail if the rate of degradation exceeds the rate of new bone formation.²¹⁻²³

In contrast, an additively manufactured fracture-tough, permanent, and biocompatible high-performance polymer scaffold with a bone-like elastic modulus has several advantages over bioresorbable and complex biomimetic scaffolds, including more predictable and tunable mechanical properties for optimal load bearing and osteoconduction. The polyaryletherketone (PAEK) family of polymers fulfills many of the requisites for SGBR, inclusive of oncological indications, with a proven safety profile and radiolucency to avoid the issues related to X-ray perturbation.¹¹ The most widely known members, polyetheretherketone (PEEK) and polyetherketoneketone (PEKK), have been employed in craniofacial implants for decades.²⁴ However, as hydrophobic and bioinert polymers, they lack the osteoconductive properties of Ti and ceramics, and so fail to osseointegrate.²⁵ We have previously shown that PEEK can be surface-modified to increase its bioactivity and osseointegration using nitrogen plasma-immersion ion implantation (PIII).²⁵⁻²⁸ In nitrogen PIII, the polymer material is immersed in nitrogen plasma (ionized nitrogen gas) and subjected to a high-voltage pulse (approximately 10keV) causing positively charged nitrogen ions to bombard and become embedded within the polymer surface. Free radicals created by the plasma treatment process form covalent bonds with adjacent proteins and substantially increase hydrophilicity and subsequent cell attachment and tissue infiltration. A less well-known member of the PAEK family, polyetherketone (PEK), is a high-performance semi-crystalline thermoplastic that is similarly fracture-tough, low-fatigue, radiolucent, able to be PIII treated, sterilizable, and has an elastic modulus better matching bone, however, there are no publications of its use in segmental bone defect repair.²⁹

Here we describe a new paradigm in patient-matched SGBR using PEK in the form of a nitrogen PIII-treated thermally toughened frame with a gyroid-based triply periodic minimal surface (TPMS) scaffold

structure, and an osteoinductive bioresorbable stem cell-infused ceramic core. Until recently, additive manufacturing techniques used on PEEK/PEKK, such as fused filament fabrication (FFF), have been unable to meet the desired print resolution or strength requirements for load-bearing implants. In contrast, PEK can be additively manufactured through the process of powder bed fusion using laser sintering (PBF-LS), which is commonly known as selective laser sintering (LS), to achieve the more complex and intricate geometries required for SGBR. LS-PEK can be toughened through annealing for improved mechanical performance and the micro-rough surface topology created through LS promotes tissue integration.³⁰ The artificial bone is custom-designed using virtual surgical planning (VSP) to match the defect site, mechanically optimized using image-based finite element (FE) analysis for strength, and numerically modelled to promote bone ingrowth. The mechanical optimization of LS-PEK provides sufficient structural stability to abrogate the need for both metal plate augmentation and autologous bone grafts. The LS-PEK frame is then dry-ice blasted for cleaning, annealed, and nitrogen PIII-treated, the latter able to withstand heat sterilization (Fig. 1).

It is conceivable that by engineering complex biomimetic structures containing multiple tissue lineages (bone, muscle, and vasculature), a more functional construct could be deployed.³¹⁻³⁴ Riffai et al. define the “Quad of tissue engineering”, comprising biomaterials, regenerative cells, morphogens/cytokines, and fabrication modality, each considered integral to repairing complex tissues such as bone. They suggest that bioink combinations with 3D-printed PCL may be structurally suitable for bone regeneration but at the same time highlighting the lack of clinically relevant scaffolds employing bioinks.³⁵ However, additional complexity encumbers clinical translation due to regulatory constraints, greater unpredictability, and the unsolved problem of how the viability of large living structures with complex geometries can be maintained when implanted. At present, this concept is better suited to tissue-engineered models for drug development. Furthermore, it is challenging to incorporate complex structures such as blood vessels into biomaterials after they have been 3D-printed into scaffolds. This makes co-printing biomaterials with bioinks the most intuitive solution. Unfortunately, the requirement for bone scaffolds to be mechanically appropriate restricts the range of biomaterials where this is feasible with many requiring high temperatures for printing or sintering that is incompatible with living tissues. To overcome this challenge, we incorporated an osteoinductive bioresorbable core that was placed within the LS-PEK frame at the time of surgery. Importantly, the ceramic core was composed of 3D-printed β TCP, designed to serve as a calcium reservoir, but non-essential for structural stability. This was infilled with gelatine methacryloyl (GelMA) encapsulating adipose derived stem cells (ADSCs) that were osteogenically pre-differentiated prior to assembly to enhance the bioavailability of β TCP. Finally, we tested our LS-PEK implant in a mature ovine segmental mandibulectomy model. Most SGBR studies employ in vivo models where the loads are considerably lower than in humans, e.g., small animals or quadruped long bones.³⁶ In contrast, the mature ovine segmental mandibulectomy model is a ‘worst case’ scenario, exaggerating the mechanical and biological challenges of SGBR in humans. Remarkably, all sheep demonstrated normal masticatory function due to osseointegration of the PIII-treated LS-PEK implant with progressive stress-driven osteoconduction through the scaffold.”

5. *In relation to the comments, the reviewer recommends that the authors refer to the two papers below and add the following papers as related references.*

(1) Kim, Joeng Ju, and Dong-Woo Cho. "Advanced strategies in 3D bioprinting for vascular tissue engineering and disease modelling using smart bioinks." Virtual and Physical Prototyping 19.1 (2024): e2395470.

(2) Rifai, Aaqil, et al. "Biofabrication of functional bone tissue: defining tissue-engineered scaffolds from nature." Frontiers in Bioengineering and Biotechnology 11 (2023): 1185841.

Response: The two publications have been added to the references. See Ref. 34 & 35

Yours Sincerely,

Professor Jonathan Clark

Professor Jeremy Micah Crook